# Sparsifying Bayesian neural networks with latent binary variables and normalizing flows

**Lars Skaaret-Lund**                                                    *lars.skaaret-lund@nmbu.no*
*Bioinformatics and Applied Statistics*
*Norwegian University of Life Sciences*

**Geir Storvik**                                                              *geirs@math.uio.no*
*Department of Mathematics*
*University of Oslo*

**Aliaksandr Hubin**                                                    *aliaksandr.hubin@nmbu.no*
*Bioinformatics and Applied Statistics*
*Norwegian University of Life Sciences*

**Reviewed on OpenReview:** *https://openreview.net/forum?id=d6kqUKzG3V*

## Abstract

Artificial neural networks are powerful machine learning methods used in many modern applications. A common issue is that they have millions or billions of parameters and tend to overfit. Bayesian neural networks (BNN) can improve on this since they incorporate parameter uncertainty. Latent binary Bayesian neural networks (LBBNN) further take into account structural uncertainty by allowing the weights to be turned on or off, enabling inference in the joint space of weights and structures. Mean-field variational inference is typically used for computation within such models. In this paper, we will consider two extensions of variational inference for the LBBNN: Firstly, by using the local reparametrization trick (LCRT), we improve computational efficiency. Secondly, and more importantly, by using normalizing flows on the variational posterior distribution of the LBBNN parameters, we learn a more flexible variational posterior than the mean field Gaussian. Experimental results on real data show that this improves predictive power compared to using mean field variational inference on the LBBNN method, while also obtaining sparser networks. We also perform a simulation study, where we consider variable selection in a logistic regression setting with highly correlated data, where the more flexible variational distribution improves results.

## 1 Introduction

Modern deep learning architectures can have billions of trainable parameters (Khan et al., 2020). Due to the large number of parameters in the model, the network can overfit, and therefore may not generalize well to unseen data. Further, the large number of parameters gives computational challenges both concerning training the network and for prediction. Various regularization methods are used to try to deal with this, such as early stopping (Prechelt, 1998), dropout (Srivastava et al., 2014) or data augmentation (Shorten & Khoshgoftaar, 2019). These techniques are heuristic and therefore it is not always clear how to use them and how well they work in practice.

Another issue with deep learning models is that they often make overconfident predictions. In Szegedy et al. (2013), it was shown that adding a small amount of noise to an image can trick a classifier into making a completely wrong prediction (with high confidence), even though the image looks the same to the human eye. The opposite is also possible, images that are white noise can be classified with almost complete certainty to belong to a specific class (Nguyen et al., 2015).

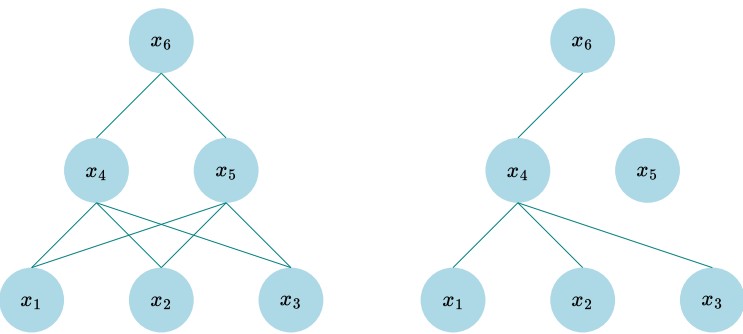

Figure 1: A dense network on the left, one possible sparse structure on the right.

Bayesian neural networks (BNN, Neal, 1992; MacKay, 1995; Bishop, 1997) use a rigorous Bayesian methodology to handle parameter and prediction uncertainty. In principle, prior knowledge can be incorporated through appropriate prior distributions, but most approaches within BNN so far only apply very simple convenience priors (Fortuin, 2022). However, approaches where knowledge-based priors are incorporated are starting to appear (Tran et al., 2022; Sam et al., 2024). Incorporating parameter uncertainty can lead to less overfitting and better-calibrated predictions, however, this comes at the expense of extremely high computational costs. Until recently, inference on Bayesian neural networks could not scale to large multivariate data due to limitations of standard Markov chain Monte Carlo (MCMC) approaches, the main quantitative procedure used for complex Bayesian inference. Recent developments of variational Bayesian approaches (Gal, 2016) allow us to approximate the posterior of interest and lead to more scalable methods.

In the frequentist setting, the lottery ticket hypothesis (Frankle & Carbin, 2018; Pensia et al., 2020) states that dense networks contain sparse subnetworks that can achieve a similar performance of the full network. These sparse subnetworks are typically obtained by threshold pruning of the weights. Similarly, in the Bayesian framework, Deng et al. (2019) obtain sparse networks with magnitude-based weight pruning. In a more recent work, Li et al. (2024) introduce a methodology where sparsity is inherent in the posterior from the very beginning of training, significantly saving on computational costs. In that work, the (non-Bayesian) inclusion parameters are also pruned, (and reintroduced back) using some pruning criteria. Another way to obtain sparse networks in the Bayesian framework is through sparsity-inducing priors. Variational dropout (Kingma et al., 2015; Molchanov et al., 2017) uses the independent log uniform prior on the weights. However, this is an improper prior, which in combination with commonly used likelihood functions leads to an improper posterior, therefore (Hron et al., 2017) argues that the obtained results can not be interpreted as a Bayesian methodology to sparsification, but rather falls into the same category as frequentist approaches. Another type of sparsity-inducing prior is the independent scale mixture prior, where Blundell et al. (2015) proposed a mixture of two Gaussian densities, where using a small variance for the second mixture component leads to many of the weights having a prior around zero.

In this work, we consider the spike-and-slab prior, used in latent binary Bayesian neural networks (LBBNN) introduced by Hubin & Storvik (2019; 2024) and concurrently in Bai et al. (2020), resulting in a formal Bayesian approach for obtaining sparse subnetworks. The spike-and-slab prior for a special case of LBBNN with the ReLu activation function was studied from a theoretical perspective in Polson & Ročková (2018). In Hubin & Storvik (2019) it was empirically shown that using this prior will induce a very sparse network (around 90 % of the weights were removed) while maintaining good predictive power. Using this approach thus takes into account uncertainty around whether each weight is included or not (structural uncertainty) and uncertainty in the included weights (parameter uncertainty) given a structure, allowing for a Bayesian approach to network sparsification (see Figure 1), thus weights are removed based on (learned) inclusion probabilities rather than ad-hoc pruning.

In Hubin & Storvik (2019; 2024), a simple mean-field variational approximation was applied for performing computation. In this paper, we show that transforming the variational posterior distribution with normalizing flows can result in even sparser networks while improving predictive power compared to the earlier work. Additionally, we demonstrate that the flow network handles predictive uncertainty well, and performs better

than the mean-field methods at variable selection in a logistic regression setting with highly correlated variables, thus demonstrating higher quality in structure learning. While the utility of LBBNN type models have been demonstrated in earlier papers (Polson & Ročková, 2018; Bai et al., 2020; Hubin & Storvik, 2019; 2024), the main contributions in this paper are:

- extending the class of mean-field variational distributions in the LBBNN context to include normalizing flows, allowing for modeling dependencies between weights while maintaining the network sparsification capabilities;

- improvements in computational efficiency and bypassing the need for continuous relaxation of binary variables in LBBNNs through the use of the local reparametrization trick (Kingma et al., 2015), by sampling the lower dimensional preactivations directly rather than the *two* much larger sets of weight and inclusion probability parameters;

- demonstrating in most cases improvements in predictive power, sparsity, and variable selection of the normalizing flow-based approximate inference through experiments on real and simulated data compared to the most relevant baseline method, LBBNN with mean-field inference;

- demonstrating robust performance in predictive uncertainty quantification across multiple problems with LBBNN producing the best results for many datasets.

## 2 The model

Given the explanatory variable $\boldsymbol{x} \in \mathbb{R}^n$, for the response variable $\boldsymbol{y} \in \mathbb{R}^m$ we assume a probability distribution $f(\cdot; \boldsymbol{\eta}(\boldsymbol{x}))$ parameterised by the vector $\boldsymbol{\eta}(\boldsymbol{x})$,

$$\boldsymbol{y} \sim f(\cdot; \boldsymbol{\eta}(\boldsymbol{x})).$$

The vector $\boldsymbol{\eta}(\boldsymbol{x})$ is obtained through a composition of semi-affine transformations:

$$u_j^{(l)} = \sigma^{(l)}\left( \sum_{i=1}^{n^{(l-1)}} u_i^{(l-1)} \gamma_{ij}^{(l)} w_{ij}^{(l)} + b_j^{(l)} \right), j = 1, \ldots, n^{(l)}, l = 1, \ldots, L, \tag{1}$$

where $\eta_j(\boldsymbol{x}) = u_j^{(L)}$, Additionally, $\boldsymbol{u}^{(l-1)}$ denotes the inputs from the previous layer (with $\boldsymbol{u}^0 = \boldsymbol{x}$ corresponding to the explanatory variables), the $w_{ij}^{(l)}$'s are the weights, the $b_j^{(l)}$'s are the bias terms, and $n^{(l)}$ (and $n^{(0)} = n$) the number of inputs at layer $l$ of a total $L$ layers. Further, we have the elementwise non-linear activation functions $\sigma^{(l)}$. The additional parameters $\gamma_{ij}^{(l)} \in \{0, 1\}$ denote binary inclusion variables for the corresponding weights.

Following Polson & Ročková (2018); Hubin & Storvik (2019); Bai et al. (2020), we consider a *structure* to be defined by the configuration of the binary vector $\boldsymbol{\gamma}$, and the weights of each structure conditional on this configuration. To consider uncertainty in both structures and weights, we use the spike-and-slab prior, where for each (independent) layer $l$ of the network, we also consider the weights to be independent:

$$p(w_{ij}^{(l)}|\gamma_{ij}^{(l)}) = \gamma_{ij}^{(l)}\mathcal{N}(w_{ij}^{(l)}; 0, (\sigma^{(l)})^2) + (1 - \gamma_{ij}^{(l)})\delta(w_{ij}^{(l)});$$
$$p(\gamma_{ij}^{(l)}) = \text{Bernoulli}(\gamma_{ij}^{(l)}; \alpha^{(l)}).$$

We will use the nomenclature from Hubin & Storvik (2019) and refer to this as the LBBNN model. Here, $\delta(\cdot)$ is the Dirac delta function, which is considered to be zero everywhere except for a spike at zero. In addition, $\sigma^2$ and $\alpha$ denote the prior variance and the prior inclusion probability of the weights, respectively. In practice, we use the same variance and inclusion probability across all the layers and weights, but this is not strictly necessary. In principle, one can incorporate knowledge about the importance of individual covariates or their co-inclusion patterns by adjusting the prior inclusion probabilities for the input layer or specifying hyper-priors. This is common in Bayesian model selection literature (Fletcher & Fletcher, 2018), but not yet within BNNs.

## 3 Bayesian inference

The main motivation behind using LBBNNs is that we are able to take into account both structural and parameter uncertainty, whereas standard BNNs are only concerned with parameter uncertainty. By doing inference through the posterior predictive distribution, we average over all possible structural configurations, and parameters. For a new observation $\tilde{\boldsymbol{y}}$ given training data, $\mathcal{D}$, we have:

$$p(\tilde{\boldsymbol{y}}|\mathcal{D}) = \sum_{\boldsymbol{\gamma}} \int_{\boldsymbol{w}} p(\tilde{\boldsymbol{y}}|\boldsymbol{w}, \boldsymbol{\gamma}, \mathcal{D}) p(\boldsymbol{w}, \boldsymbol{\gamma}|\mathcal{D}) \, d\boldsymbol{w}.$$

This expression is intractable due to the ultra-high dimensionality of $\boldsymbol{w}$ and $\boldsymbol{\gamma}$, and using Monte Carlo sampling as an approximation is also challenging due to the difficulty of obtaining samples from the posterior distribution, $p(\boldsymbol{w}, \boldsymbol{\gamma}|\mathcal{D})$. Instead of trying to sample from the true posterior, we turn it into an optimization problem, using variational inference (VI, Blei et al., 2017). The key idea is that we replace the true posterior distribution with an approximation, $q_{\boldsymbol{\theta}}(\boldsymbol{w}, \boldsymbol{\gamma})$, with $\boldsymbol{\theta}$ denoting some variational parameters. We learn the variational parameters that make the approximate posterior as close as possible to the true posterior. Closeness is measured through the Kullback-Leibler (KL) divergence,

$$\text{KL}\left[q_{\boldsymbol{\theta}}(\boldsymbol{w}, \boldsymbol{\gamma}) || p(\boldsymbol{w}, \boldsymbol{\gamma}|\mathcal{D})\right] = \sum_{\boldsymbol{\gamma}} \int_{\boldsymbol{w}} q_{\boldsymbol{\theta}}(\boldsymbol{w}, \boldsymbol{\gamma}) \log \frac{q_{\boldsymbol{\theta}}(\boldsymbol{w}, \boldsymbol{\gamma})}{p(\boldsymbol{w}, \boldsymbol{\gamma}|\mathcal{D})} \, d\boldsymbol{w}. \tag{2}$$

Minimizing the KL-divergence (with respect to $\boldsymbol{\theta}$) is equivalent to maximizing the evidence lower bound (ELBO):

$$\text{ELBO}(q_{\boldsymbol{\theta}}) = \mathbb{E}_{q_{\boldsymbol{\theta}}(\boldsymbol{w}, \boldsymbol{\gamma})}\left[\log p(\mathcal{D}|\boldsymbol{w}, \boldsymbol{\gamma})\right] - \text{KL}\left[q_{\boldsymbol{\theta}}(\boldsymbol{w}, \boldsymbol{\gamma}) || p(\boldsymbol{w}, \boldsymbol{\gamma})\right]. \tag{3}$$

The objective is thus to maximize the expected log-likelihood while penalizing with respect to the KL divergence between the prior and the variational posterior. How good the approximation becomes depends on the family of variational distributions $\{q_{\boldsymbol{\theta}}, \boldsymbol{\theta} \in \Theta\}$ that is chosen.

### 3.1 Choices of variational families

A common choice (Blundell et al., 2015) for the approximate posterior in (dense) Bayesian neural networks is the mean-field Gaussian distribution. For simplicity of notation, denote now by $\mathbf{W}$ the set of weights corresponding to a specific layer. Note that from here on, we drop the layer notation for readability, since the parameters at different layers will always be considered independent in both the variational distribution and the prior. Then

$$q_{\boldsymbol{\theta}}(\mathbf{W}) = \prod_{i=1}^{n_{in}} \prod_{j=1}^{n_{out}} \mathcal{N}(w_{ij}; \tilde{\mu}_{ij}, \tilde{\sigma}_{ij}^2),$$

where $n_{in}$ and $n_{out}$ denote the number of neurons in the previous and current layer, respectively. Weights corresponding to different layers are assumed independent as well. The mean-field Gaussian distribution for Bayesian neural networks can be extended to include the binary inclusion variables following Carbonetto & Stephens (2012):

$$q_{\boldsymbol{\theta}}(\mathbf{W}|\boldsymbol{\Gamma}) = \prod_{i=1}^{n_{in}} \prod_{j=1}^{n_{out}} \left(\gamma_{ij}\mathcal{N}(w_{ij}; \tilde{\mu}_{ij}, \tilde{\sigma}_{ij}^2) + (1 - \gamma_{ij})\delta(w_{ij})\right);$$

$$q_{\tilde{\alpha}_{ij}}(\gamma_{ij}) = \text{Bernoulli}(\gamma_{ij}; \tilde{\alpha}_{ij}). \tag{4}$$

Here, $\boldsymbol{\Gamma}$ is the set of inclusion indicators corresponding to a specific layer. However, the mean-field Gaussian distribution (Blundell et al., 2015) is typically too simple to be able to capture the complexity of the true posterior distribution. We follow Ranganath et al. (2016), and introduce a set of auxiliary latent variables $\boldsymbol{z}$ to model dependencies between the weights in $q$, and use the following variational posterior distribution:

$$q_{\boldsymbol{\theta}}(\mathbf{W}|\boldsymbol{\Gamma}, \boldsymbol{z}) = \prod_{i=1}^{n_{in}} \prod_{j=1}^{n_{out}} \left(\gamma_{ij}\mathcal{N}(w_{ij}; z_i\tilde{\mu}_{ij}, \tilde{\sigma}_{ij}^2) + (1 - \gamma_{ij})\delta(w_{ij})\right);$$

$$q_{\tilde{\alpha}_{ij}}(\gamma_{ij}) = \text{Bernoulli}(\gamma_{ij}; \tilde{\alpha}_{ij}), \tag{5}$$

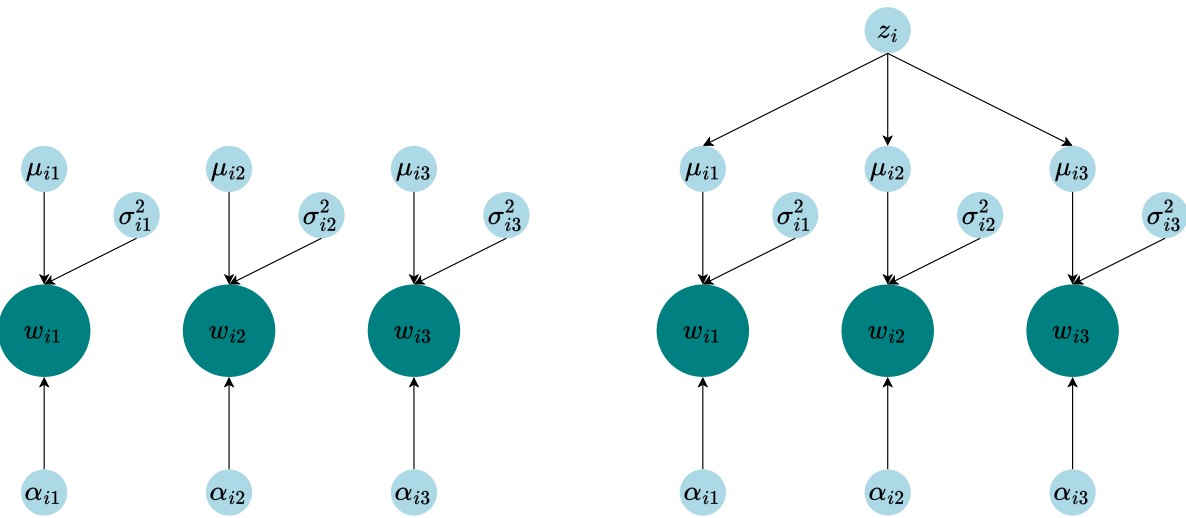

Figure 2: On the left, the mean-field variational posterior where the weights are assumed independent. On the right, the latent variational distribution $z$ allows for modeling dependencies between the weights.

where $\boldsymbol{z} = (z_1, ..., z_{n_{in}})$ follows a distribution $q_\phi(\boldsymbol{z})$. For an illustration of the difference between the two variational distributions in Equation (4) and Equation (5), see Figure 2. The novelty in our suggested variational distribution is to combine both weight and structural uncertainty, in addition to modeling dependencies between the weights. As for $\mathbf{W}$, also $\boldsymbol{z}$ is a set of variables related to a specific layer and independence between layers is assumed also for $\boldsymbol{z}$'s. To increase the flexibility of the variational posterior, we apply normalizing flows (Rezende & Mohamed, 2015) to $q_\phi(\boldsymbol{z})$. In general, a normalizing flow is a composition of invertible transformations of some initial (simple) random variable $\mathbf{z}_0$,

$$\boldsymbol{z}_k = f_k(\boldsymbol{z}_{k-1}), \quad k = 1, ..., K.$$

The log density of the transformed variable $\boldsymbol{z} = \boldsymbol{z}_K$ is given as,

$$\log q(\boldsymbol{z}) = \log q_0(\boldsymbol{z}_0) - \sum_{k=1}^{K} \log \left| \det \frac{\partial \boldsymbol{z}_k}{\partial \boldsymbol{z}_{k-1}} \right|. \tag{6}$$

We are typically interested in transformations that have a Jacobian determinant that is tractable, and fast to compute, in addition to being highly flexible. Transforming the variational posterior distribution in a BNN with normalizing flows was first done in Louizos & Welling (2017), who coined the term multiplicative normalizing flows (BNN-FLOW), where the transformations were applied in the activation space instead of the weight space. As the weights are of much higher dimensions, the number of flow parameters and thus the number of parameters of variational distribution would explode quickly. We will follow Louizos & Welling (2017) here. The main difference in our work is that by using the variational posterior in Equation (5), we also get sparse networks.

For the normalizing flows, we will use the inverse autoregressive flow (IAF), with numerically stable updates, introduced by Kingma et al. (2016). It works by transforming the input in the following way:

$$\begin{aligned}
\boldsymbol{z}_{k-1} &= \text{input} \\
\boldsymbol{m}_k, \boldsymbol{s}_k &= g(\boldsymbol{z}_{k-1}) \\
\boldsymbol{\kappa}_k &= \text{sigmoid}(\boldsymbol{s}_k) \\
\boldsymbol{z}_k &= \boldsymbol{\kappa}_k \odot \boldsymbol{z}_{k-1} + (1 - \boldsymbol{\kappa}_k) \odot \boldsymbol{m}_k,
\end{aligned} \tag{7}$$

where $g$ is a neural network and $\odot$ denotes elementwise multiplication. Assuming the neural network in Equation (7) is autoregressive (i.e $z_{k,i}$ can only depend on $z_{k,1:i-1}$), we get a lower triangular Jacobian and

$$\log \left| \det \frac{\partial \boldsymbol{z}_k}{\partial \boldsymbol{z}_{k-1}} \right| = \sum_{i=1}^{n_{in}} \log \kappa_{k,i}. \tag{8}$$

## 3.2 Computing the variational bounds

Minimization of the KL in Equation (2) is difficult due to the introduction of the auxiliary variable $\boldsymbol{z}$ in the variational distribution. In principle, $\boldsymbol{z}$ could be integrated out, but in practice this is difficult. Following Ranganath et al. (2016), we instead introduce $\boldsymbol{z}$ as an auxiliary variable also in the posterior distribution by defining

$$p(\boldsymbol{w}, \boldsymbol{\gamma}, \boldsymbol{z}|\mathcal{D}) = p(\boldsymbol{w}, \boldsymbol{\gamma}|\mathcal{D})r(\boldsymbol{z}|\boldsymbol{w}, \boldsymbol{\gamma})$$

where $r(\boldsymbol{z}|\boldsymbol{w}, \boldsymbol{\gamma})$ in principle can be any distribution. We then consider the KL divergence in the extended space for $(\boldsymbol{w}, \boldsymbol{\gamma}, \boldsymbol{z})$:

$$\mathrm{KL}\left[q(\boldsymbol{w}, \boldsymbol{\gamma}, \boldsymbol{z})||p(\boldsymbol{w}, \boldsymbol{\gamma}, \boldsymbol{z}|\mathcal{D})\right] = \int_{\boldsymbol{z}} \sum_{\boldsymbol{\gamma}} \int_{\boldsymbol{w}} q(\boldsymbol{w}, \boldsymbol{\gamma}, \boldsymbol{z}) \log \frac{q(\boldsymbol{w}, \boldsymbol{\gamma}, \boldsymbol{z})}{p(\boldsymbol{w}, \boldsymbol{\gamma}, \boldsymbol{z}|\mathcal{D})} \, d\boldsymbol{w} d\boldsymbol{z}$$

which, by utilizing the definitions of $p(\boldsymbol{w}, \boldsymbol{\gamma}, \boldsymbol{z})$ and $q(\boldsymbol{w}, \boldsymbol{\gamma}, \boldsymbol{z})$ can be rewritten to

$$\mathrm{KL}\left[q(\boldsymbol{w}, \boldsymbol{\gamma}, \boldsymbol{z})||p(\boldsymbol{w}, \boldsymbol{\gamma}, \boldsymbol{z}|\mathcal{D})\right]$$
$$= \mathbb{E}_{q(\boldsymbol{z})}\left[\mathrm{KL}\left[q(\boldsymbol{w}, \boldsymbol{\gamma}|\boldsymbol{z})||p(\boldsymbol{w}, \boldsymbol{\gamma})\right] + \log q(\boldsymbol{z})\right] - \mathbb{E}_{q(\boldsymbol{W}, \boldsymbol{\Gamma}, \boldsymbol{z})}\left[\log p(\mathcal{D}|\boldsymbol{w}, \boldsymbol{\gamma}) + \log r(\boldsymbol{z}|\boldsymbol{w}, \boldsymbol{\gamma})\right] + \log p(\mathcal{D}). \tag{9}$$

It follows directly from the chain rule for relative entropies (see Cover, 1999, Theorem 2.5.3) that

$$\mathrm{KL}\left[q(\boldsymbol{w}, \boldsymbol{\gamma})||p(\boldsymbol{w}, \boldsymbol{\gamma}|\mathcal{D})\right] \leq \mathrm{KL}\left[q(\boldsymbol{w}, \boldsymbol{\gamma}, \boldsymbol{z})||p(\boldsymbol{w}, \boldsymbol{\gamma}, \boldsymbol{z}|\mathcal{D})\right], \tag{10}$$

giving a looser than the original upper bound, but the dependence structure in the variational posterior distribution can compensate for this.

After doing some algebra, we get the following contribution to the first term within the first expectation in Equation (9) from a specific layer:

$$\mathrm{KL}\left[q(\boldsymbol{w}, \boldsymbol{\gamma}|\boldsymbol{z})||p(\boldsymbol{w}, \boldsymbol{\gamma})\right] = \sum_{ij} \left( \tilde{\alpha}_{ij} \left( \log \frac{\sigma_{ij}}{\tilde{\sigma}_{ij}} + \log \frac{\tilde{\alpha}_{ij}}{\alpha_{ij}} - \frac{1}{2} + \frac{\tilde{\sigma}_{ij}^2 + (\tilde{\mu}_{ij} z_i - 0)^2}{2\sigma_{ij}^2} \right) + (1 - \tilde{\alpha}_{ij}) \log \frac{1 - \tilde{\alpha}_{ij}}{1 - \alpha_{ij}} \right).$$

Since we use autoregressive flows, the contribution to the second term in the first expectation simplifies to

$$\log q(\boldsymbol{z}) = \log q_0(\boldsymbol{z}_0) - \sum_{k=1}^{K} \sum_{i=1}^{n_{in}} \log \kappa_{k,i}.$$

For the specific choice of $r(\boldsymbol{z}|\boldsymbol{w}, \boldsymbol{\gamma})$, we follow Louizos et al. (2017) in choosing

$$r_B(\boldsymbol{z}_B|\boldsymbol{w}, \boldsymbol{\gamma}) = \prod_{i=1}^{n_{in}} \mathcal{N}(\nu_i, \tau_i^2).$$

We define the dependence of $\boldsymbol{\nu} = (\nu_1, ..., \nu_{in})$ and $\boldsymbol{\tau}^2 = (\tau_1^2, ..., \tau_{in}^2)$ on $\boldsymbol{w}$ and $\boldsymbol{\gamma}$ similar to Louizos & Welling (2017):

$$\begin{aligned} \boldsymbol{\nu} &= n_{\mathrm{out}}^{-1}(\boldsymbol{d}_1 \boldsymbol{s}^T)\boldsymbol{1}, && \text{with } \boldsymbol{s} = \zeta(\mathbf{e}^T(\boldsymbol{w} \odot \boldsymbol{\gamma})) \\ \log \boldsymbol{\tau}^2 &= n_{\mathrm{out}}^{-1}(\boldsymbol{d}_2 \boldsymbol{s}^T)\boldsymbol{1}. \end{aligned} \tag{11}$$

Here, $\boldsymbol{d}_1$, $\boldsymbol{d}_2$ and $\mathbf{e}$ are trainable parameters with the same shape as $\boldsymbol{z}$. For $\zeta$, we use hard-tanh[1], as opposed to tanh (used in Louizos & Welling (2017)) as this works better empirically. For the second last term of Equation (9), we thus have:

$$\log r\left(\boldsymbol{z}|\mathbf{w},\boldsymbol{\gamma}\right) = \log r_B\left(\boldsymbol{z}_B|\mathbf{w},\boldsymbol{\gamma}\right) + \log\left|\det\frac{\partial\boldsymbol{z}_B}{\partial\boldsymbol{z}}\right|.$$

This means that we must use two normalizing flows, one to get from $\boldsymbol{z}_0$ to $\boldsymbol{z} = \boldsymbol{z}_K$, and another from $\boldsymbol{z}$ to $\boldsymbol{z}_B$. Here, we have used the inverse normalizing flow of the form (7) with only one layer, but this can, in general, be extended to an arbitrary number of them just like in Equation (6).

For the biases of a given layer, we assume they are independent of the weights, and each other. We use the standard normal prior with the mean-field Gaussian approximate posterior. As we do not use normalizing flows on the biases, we only need to compute the KL-divergence between two Gaussian distributions:

$$\text{KL}\left[q(\boldsymbol{b})||p(\boldsymbol{b})\right] = \sum_{ij}\left(\log\frac{\sigma_{b_{ij}}}{\tilde{\sigma}_{b_{ij}}} - \frac{1}{2} + \frac{\tilde{\sigma}_{b_{ij}}^2 + (\tilde{\mu}_{b_{ij}} - 0)^2}{2\sigma_{b_{ij}}^2}\right).$$

In practice, the ELBO is optimized through a (stochastic) gradient algorithm where the reparametrization trick (Kingma & Welling, 2013) combined with mini-batch is applied. This involves sampling the large $\Gamma$ and $\mathbf{W}$ matrices.

## 4  Combining LBBNNs with the LCRT and MNF

We use an approximation combining the local parametrization trick (LCRT, Kingma et al., 2015) with the multiplicative normalizing flow method. This means sampling is based on the pre-activations $h_j = b_j + \sum_{i=1}^{N} o_i z_i \gamma_{ij} w_{ij}$ (where $\{o_i\}$ denotes the outputs from the previous layer). The mean and the variance of the activation $h_j$ are:

$$\mathbb{E}[h_j] = \tilde{\mu}_{b_j} + \sum_{i=1}^{N} o_i z_i \tilde{\alpha}_{ij} \tilde{\mu}_{ij}$$

$$\text{Var}[h_j] = \tilde{\sigma}_{b_j}^2 + \sum_{i=1}^{N} o_i^2 \tilde{\alpha}_{ij}(\tilde{\sigma}_{ij}^2 + (1 - \tilde{\alpha}_{ij})z_i^2\tilde{\mu}_{ij}^2).$$

It should be noted that affects both the mean and the variance of our Gaussian approximation, whereas in Louizos & Welling (2017) it only influences the mean. Louizos & Welling (2017) also sample one $\boldsymbol{z}$ for each observation within the mini-batch. We found that empirically it made no difference in performance to only sample one vector and multiply the same $\boldsymbol{z}$ with each input vector. We do this, as it is more computationally efficient.

A simplified version only applying the LCRT method assumes $z_i = 1$ for all $i$. We will denote this method by LBBNN-LCRT in section 5. The method combining both LCRT and MNF will be denoted LBBNN-FLOW.

In addition to a huge reduction in the number of variables to be sampled, we get a reduction in the variance of the gradient estimates, as shown in Kingma et al. (2015). Note also that the approximations induced by the sampling procedure for $\boldsymbol{h}$ can be considered as an alternative variational approximation directly for $p(\boldsymbol{h}|\mathcal{D})$.

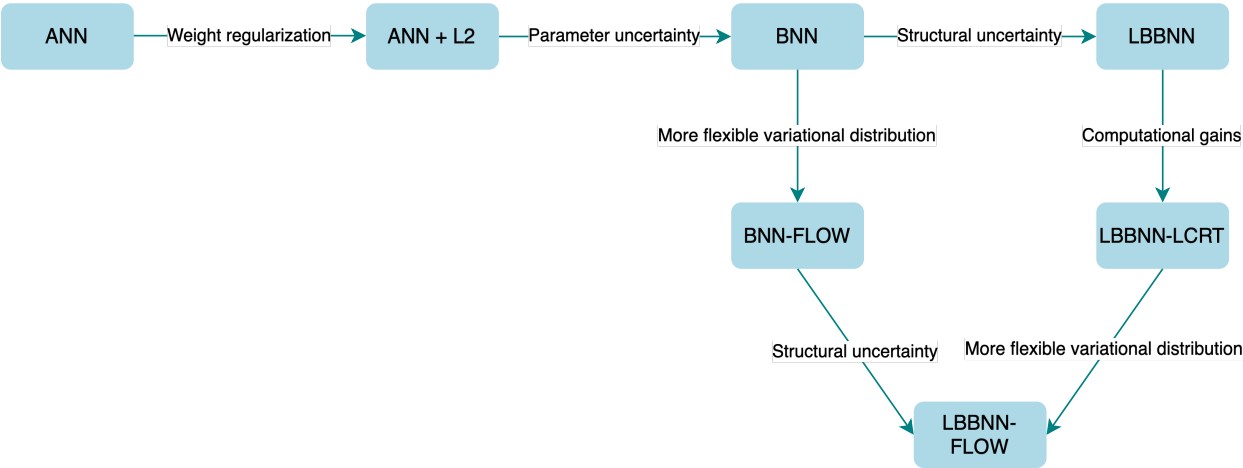

Figure 3: Illustration of the relations between the different methods considered in this paper. Exactly one design change is present between all direct neighbors disregarding the directions of the edges, thus allowing us to interpret the edges as treatments between specific pairs of methods.

## 5 Experiments

In this section, we demonstrate the robustness of our approach and show improvements with respect to the closest baseline methods of Hubin & Storvik (2024), (denoted LBBNN-SSP-MF in their paper), denoted LBBNN here, with the two approaches proposed in this paper. We consider both Bayesian model averaging (Hoeting et al., 1999) over the variational posterior distribution for the inclusion variables and weights, as well as the use of the median probability model (Barbieri et al., 2018), only sampling from weights with posterior probabilities for inclusion variables above 0.5. Median probability models have the potential to give significant sparsity gains. We also compare it to other reasonable baseline methods that allow us to evaluate the direct treatment effects. Our aim is to demonstrate the potential for improvements by the use of the proposed method, focusing on comparison under similar settings. For all methods, further improvements are possible through extensive tuning of hyper-parameters and different ad-hoc tricks commonly found in the Bayesian deep learning literature that are utilized to improve on predictive performance (e.g. tempering the posterior (Wenzel et al., 2020) or clipping the variance of the variational posterior distribution as done in Louizos & Welling (2017)). Using these tricks (although tempting) would not allow us to evaluate the pure contribution of the methodology. For the classification experiments, we provide comparisons to a standard frequentist neural network (ANN) without regularization, corresponding to the maximum likelihood estimation of the network weights, in addition to one with L2 regularization (ANN + L2), corresponding to the maximum a posteriori estimator (MAP) with independent Gaussian priors from a standard Bayesian neural network (BNN). We also have a standard BNN, taking weight uncertainty into account. From there, we get to the LBBNN method by having an extra inclusion parameter per weight, allowing for a sparse BNN. For LBBNN, exactly the same parameter priors (slab components) as in BNN were used, allowing us to evaluate the effects of adding the structural uncertainty. The multiplicative normalizing flow method (BNN-FLOW) is also closely related to a standard BNN, but here instead of sparsifying the network, we allow the variational posterior distribution to be more flexible than a standard mean-field Gaussian, used in the BNN. Further, using the local reparametrization trick (LBBNN-LCRT) is mainly a computational advantage compared to the LBBNN method. Finally, LBBNN-FLOW (proposed in this paper) is related to *both* BNN-FLOW and LBBNN-LCRT, in the sense that it can learn a sparse BNN, and in addition, have a more flexible posterior distribution than the mean-field Gaussian used in LBBNN-LCRT. For a graphical illustration of how these methods are related, we refer to Figure 3.

---

[1]hard-tanh(x) = $\begin{cases} -1, & x < -1 \\ x, & -1 \leq x \leq 1 \\ 1, & x > 1 \end{cases}$

Table 1: Performance metrics on the logistic regression variable selection simulation study.

|  | CS | LBBNN-LCRT | LBBNN-FLOW |
|---|---|---|---|
| mean TPR | 0.681 | 0.838 | **0.972** |
| mean FPR | 0.125 | 0.084 | **0.074** |

In addition to the classification experiments in the main text, we also provide experiments (both classification and regression) on various tabular datasets in Appendix D. Here, in addition to the methods detailed above, we also include Monte Carlo (MC) dropout (Gal & Ghahramani, 2016), and (exact) Gaussian processes (more details about how this is implemented in Appendix D) for the regression datasets. Finally, for the tabular datasets, we check the internal parsimony of all of the Bayesian baselines through the $p_{\text{WAIC}_1}$ and $p_{\text{WAIC}_2}$ penalties from Gelman et al. (2014), metrics that also have been used as estimates of the effective number of parameters.

Additionally, we perform two simulation studies. In the first one, we consider variable selection in a logistic regression setting, with highly correlated explanatory variables. We further vary correlation levels and noise levels of linear predictor. In additional experiments reported in Appendix B, we address 3 different correlation structures and relations to the response, where we also change the level of correlations between the covariates and repeat that for several noise levels.

All the experiments with LBBNN were coded in Python, using the PyTorch deep learning library (Paszke et al., 2019), but the approach of Carbonetto & Stephens (2012) was run in R.

## 5.1 Logistic regression simulation study

In this section, we do a variable selection experiment within a logistic regression setting. As logistic regression is just a special case of a neural network with one neuron (and hence one layer), modifying the algorithms is straightforward. We are limiting ourselves to the logistic regression context to be able to compare to the original baseline method from Carbonetto & Stephens (2012), who demonstrated that the mean-field variational approximation starts to fail the variable selection task when the covariates are correlated. As we are only interested in comparing the mean-field variational approach against the variational distribution with normalizing flows, we do not include comparisons with more traditional variable selection methods such as Lasso (Tibshirani, 1996) or Elastic Net (Zou & Hastie, 2005).

We use the same data as in Hubin & Storvik (2018), consisting of a mix of 20 binary and continuous variables, with a binary outcome, and 2 000 observations. The covariates, $\boldsymbol{x}$, are generated with a strong and complicated correlation structure between many of the variables (see Figure 4). For more details on exactly how the covariates are generated, see appendix B of Hubin & Storvik (2018). The response variable, $y$, is generated according to the following data-generating process:

$$\eta \sim \mathcal{N}(\boldsymbol{\beta}^T \boldsymbol{x}, 0.5)$$
$$y \sim \text{Bernoulli}\left(\frac{\exp(\eta)}{1 + \exp(\eta)}\right) \tag{12}$$

with the regression parameters defined to be:

$$\boldsymbol{\beta}^T = (-4, 0, 1, 0, 0, 0, 1, 0, 0, 0, 1.2, 0, 37.1, 0, 0, 50, -0.00005, 10, 3, 0).$$

The goal is to train the different methods to select the non-zero elements of $\boldsymbol{\beta}$. We consider the parameter $\beta_j$ to be included if the corresponding posterior inclusion probability $\tilde{\alpha}_j > 0.5$, i.e. the median probability model of Barbieri & Berger (2004). We fit the different methods 100 times (to the same data), each time computing the true positive rate (TPR), and the false positive rate (FPR).

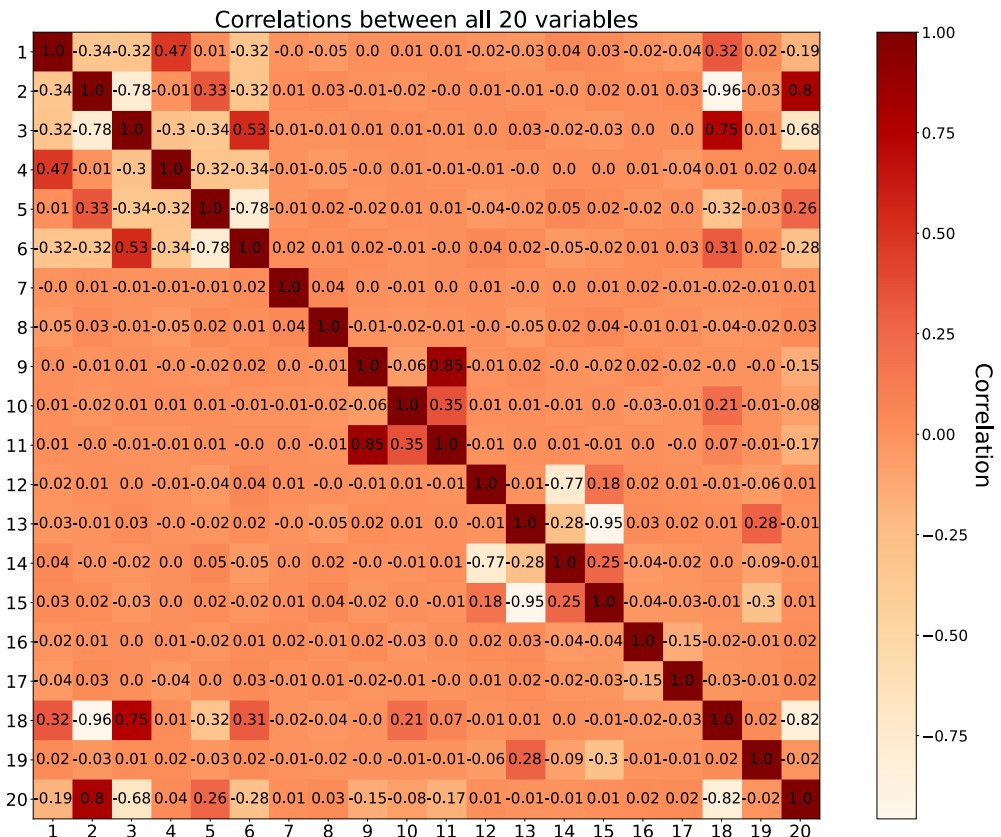

Figure 4: Plots showing the correlation between different variables in the logistic regression simulation study.

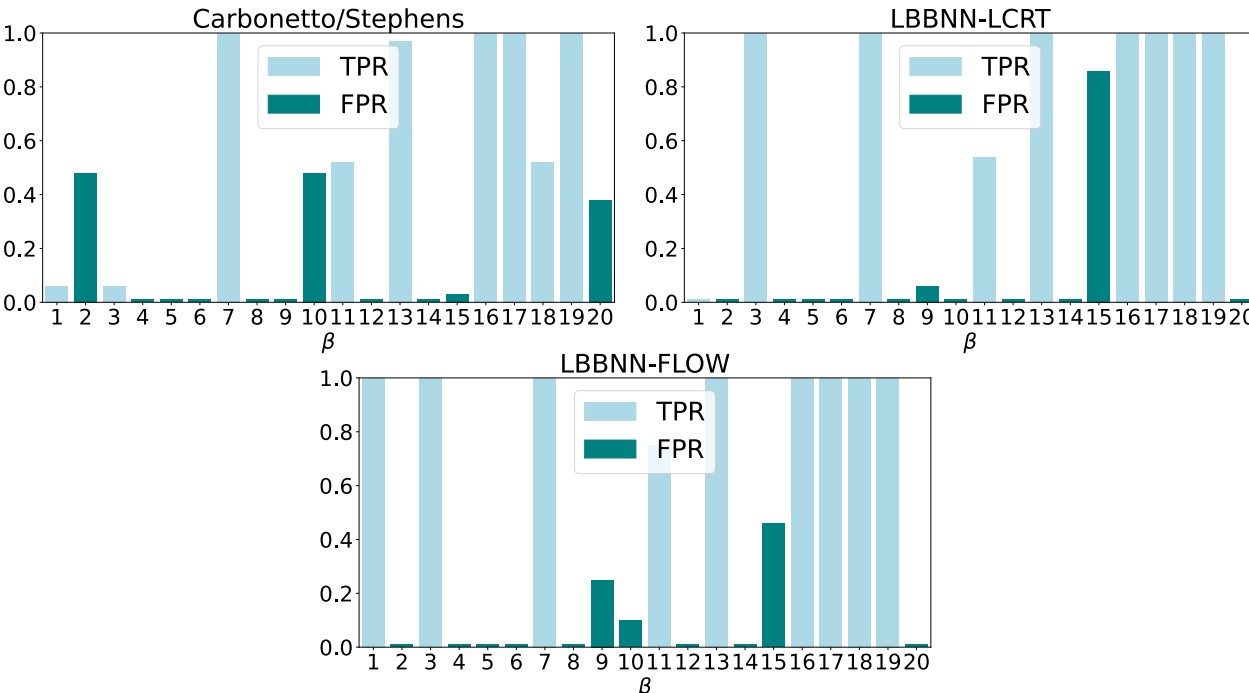

Figure 5: Bar-plots showing how often the weights are included over 100 runs.

In this experiment we compare our approaches LBBNN-LCRT and LBBNN-FLOW against the algorithm proposed by Carbonetto & Stephens (2012), denoted as CS henceforth. That method is very similar to LBBNN-LCRT, as it uses the same variational distribution. But in CS, optimization is done with coordinate

ascent variational inference and without subsampling from the data. For the normalizing flows, we use flows of length two with the neural networks having two hidden layers of 100 neurons each. We use a batch size of 400 and train for 500 epochs. We use standard normal priors for the weights and a prior inclusion probability of 0.25 on the inclusion indicators for all three approaches. Hence, we are in the setting of a Bayesian logistic regression, with variable selection.

The results are in Table 1. We also show a bar-plot (Figure 5) for each of the 20 weights over the 100 runs. We see that LBBNN-FLOW performs best, with the highest TPR and the lowest FPR. It is especially good at picking out the correct variables where there is a high correlation between many of them (for example $\beta_1 - \beta_6$). We might attribute this to the more flexible variational posterior distribution, as opposed to the mean-field Gaussian distribution used in the other three methods. Carbonetto & Stephens (2012) also discuss how the mean-field approach can only be expected to be a good approximation when the variables are independent or at most weakly correlated.

### 5.1.1 Extension to different levels of correlation

To extend the experiment to other correlation levels, we use the same data as in the simulation study from Hubin & Storvik (2018), but with varying noise levels, using $\sigma \in \{1, 0.1, 0.01\}$ that we use to generate $\eta$ and with 10 different correlation structures for $x_{i,j}$ per signal level. For a correlation parameter $\phi \in \{1, 0.9, \ldots, 0.1\}$, the covariates were updated by the following rule: $x_{i,j} = \rho \cdot x_{i,j-4} + (1 - \rho) \cdot x_{i,j}, i = 1, ..., 2000, j = 5, ..., 20$. Then, the resulting predictors are standardized resulting in the final correlated covariates used to generate the responses just like described in equation 12 above. Finally, CS, LBBNN-LCRT, and LBBNN-FLOW were run 20 times per setting using exactly the same hyper and tuning parameters as in the previous simulation study.

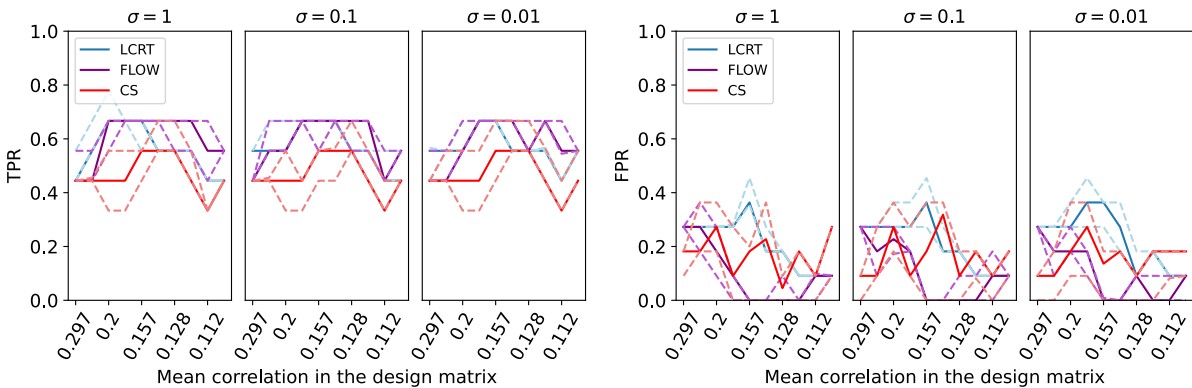

Figure 6: On the left, true positive rates, with false positive rates to the right. The maximum correlation ranged between 0.954 and 0.997.

For this experiment, in Figure 6, we show plots of average true positive and false positive rates, where the dashed lines indicate a 90% empirical confidence interval obtained from 20 runs of the corresponding algorithms per settings on different seeds. Along the x-axis, we show (every other one to save space) the mean absolute correlation between all covariates in the corresponding setting. As mentioned previously, for this experiment we consider 30 different settings, divided into 3 parts where we simulated three different noise levels in the linear predictor, and for each of these, we have 10 different correlation levels. The results in Figure 6 are consistent with those obtained on the original data from Hubin & Storvik (2018). Across all settings here, the LBBNN-FLOW is on average the most favorable method with respect to a trade-off between TPR and FPR, although in many cases the confidence intervals are overlapping.

We added three more studies with other complicated correlation structures, varying signal strength (3 different levels in every study), and correlation levels (10 levels per signal level) between the covariates. These additional simulation studies are provided in the Appendix B to this paper.

Other results with varying levels of correlations between one of the data generative covariates and one of the non-data generative covariates are reported in follow-up works (Sommerfelt & Hubin, 2024; Høyheim, 2024) that use our approaches. There, in all scenarios, both Sommerfelt & Hubin (2024) and Høyheim (2024) show that the flow-based inference is outperforming the LCRT one.

## 5.2 Classification experiments

### 5.2.1 Experimental setup

We perform two classification experiments, one with the same fully connected architecture as in Hubin & Storvik (2019), and the other with a convolutional architecture (see appendix C for details on how this is implemented). On the fully connected architecture, we use two hidden layers with 400 and 600 neurons respectively, whereas, for the other experiment, we use the LeNet-5 (LeCun et al., 1998) convolutional architecture, but with 32 and 48 filters for the convolutional layers. We emphasize that it is possible to use deeper and more complicated architectures (for example Resnet-18, He et al., 2016), which would likely improve the results reported in this paper. As the goal here is not to try to approach (or hack through tuning and engineering) state-of-the-art results, we do not experiment any further with this. In both cases, we classify on MNIST (Deng, 2012), FMNIST (Fashion MNIST) (Xiao et al., 2017) and KMNIST (Kuzushiji MNIST) (Clanuwat et al., 2018). All of these datasets contain 28x28 grayscale images, divided into a training and validation set with $60\,000$ and $10\,000$ images respectively. MNIST and FMNIST are well-known and often utilized datasets, so it is easy to compare performance when testing novel algorithms, while KMNIST is a more recent addition.

For both the fully connected and convolutional architectures, we use the Adam (Kingma & Ba, 2014) optimizer, batch size of 100, and train for 250 epochs. All the experiments are run 10 times, and we report the minimum, median, and maximum predictive accuracy over these 10 runs as well as expected calibration errors and negative test log-likelihood. We measure predictive performance and predictive uncertainty handling within two approaches: First, the Bayesian model averaging approach, where we average over 100 samples from the variational posterior distribution, where for our proposed method we are taking into account uncertainty in both weights and structures following Hubin & Storvik (2019). For the standard BNN and BNN-FLOW, this model averaging is only over the weights, whereas for the frequentist approaches, we do not have any model averaging. Secondly, for the methods with both weight and structural uncertainty, we consider the median probability model (Barbieri & Berger, 2004), where we only do averaging over the weights that have a posterior inclusion probability greater than 0.5, whilst others are excluded from the model. This allows for significant sparsification of the network. We emphasize that this is possible because we can go back to sampling the weights when doing inference, i.e. we sample only from the weights that have a corresponding inclusion probability greater than 0.5.

In addition (where applicable), we also report the density of the network, defined as the proportion of non-zero weights. The reported density (1-sparsity) is an average over these 10 runs. For the full variational model averaging approach, we consider the density to be equal to one since we do not explicitly exclude any weights when computing the predictions (even though a large proportion of the weights may have a small inclusion probability and in practice within any 10 samples over which we are marginalizing less than 100% of the weights will be used, yet ideally one wants to average over more than 10 samples).

### 5.2.2 Methods

For these experiments, we are interested in comparing our method LBBNN-FLOW against the closest baseline methods (see Figure 3). For ANN + L2, we use weight decay of 0.5, inducing a penalized likelihood, which corresponds to MAP (maximum aposteriori probability) solutions of BNN and BNN-FLOW under standard normal priors. For BNN and BNN-FLOW, we use standard normal priors for the weights. For the LBBNN-LCRT and LBBNN-FLOW methods, we also use the standard normal prior for the slab components of all the weights and biases in the network, and a prior inclusion probability of 0.10. For both $q(\boldsymbol{z})$ and $r(\boldsymbol{z}|\mathbf{W},\boldsymbol{\Gamma})$, we use flows of length two, where the neural networks consist of two hidden layers with 250 neurons each. For the Bayesian model averaging approaches, we average over 100 samples from the variational posterior

Table 2: Performance metrics (accuracy and density) on the KMNIST, MNIST, FMNIST validation data, for the fully connected architecture. For the accuracies (%), we report the minimum, maximum, and median over the ten different runs. Density is computed as an average over the ten runs. The best median results are bold.

| KMNIST | Median probability model | | | | Dense model | | | |
|---|---|---|---|---|---|---|---|---|
| Method | min | median | max | density | min | median | max | density |
| LBBNN | 89.34 | 89.69 | 90.19 | 0.113 | 89.62 | 89.87 | 90.29 | 1.000 |
| LBBNN-LCRT | 89.94 | 90.21 | 90.42 | 0.136 | 90.17 | 90.35 | 90.62 | 1.000 |
| LBBNN-FLOW | 90.89 | **91.04** | 91.37 | **0.096** | 91.23 | 91.43 | 91.65 | 1.000 |
| BNN-FLOW | - | - | - | - | 91.86 | 92.33 | 92.74 | 1.000 |
| BNN | - | - | - | - | 92.29 | **92.47** | 92.66 | 1.000 |
| ANN | - | - | - | - | 89.61 | 90.82 | 91.26 | 1.000 |
| ANN + L2 | - | - | - | - | 87.20 | 87.84 | 88.22 | 1.000 |
| **MNIST** | Median probability model | | | | Dense model | | | |
| Method | min | median | max | density | min | median | max | density |
| LBBNN | 97.99 | 98.07 | 98.14 | 0.098 | 97.99 | 98.07 | 98.18 | 1.000 |
| LBBNN-LCRT | 97.83 | 97.98 | 98.05 | 0.103 | 98.00 | 98.10 | 98.20 | 1.000 |
| LBBNN-FLOW | 98.23 | **98.38** | 98.52 | **0.074** | 98.33 | 98.43 | 98.50 | 1.000 |
| BNN-FLOW | - | - | - | - | 98.39 | **98.54** | 98.67 | 1.000 |
| BNN | - | - | - | - | 98.40 | 98.48 | 98.58 | 1.000 |
| ANN | - | - | - | - | 97.98 | 98.10 | 98.18 | 1.000 |
| ANN + L2 | - | - | - | - | 96.95 | 97.04 | 97.16 | 1.000 |
| **FMNIST** | Median probability model | | | | Dense model | | | |
| Method | min | median | max | density | min | median | max | density |
| LBBNN | 88.56 | 88.84 | 89.04 | 0.107 | 88.66 | 88.84 | 89.16 | 1.000 |
| LBBNN-LCRT | 87.44 | 87.76 | 88.05 | 0.141 | 87.76 | 87.98 | 88.05 | 1.000 |
| LBBNN-FLOW | 89.42 | **89.65** | 89.91 | **0.097** | 89.47 | 89.80 | 90.03 | 1.000 |
| BNN-FLOW | - | - | - | - | 89.05 | 89.44 | 89.62 | 1.000 |
| BNN | - | - | - | - | 90.02 | **90.17** | 90.37 | 1.000 |
| ANN | - | - | - | - | 89.31 | 89.58 | 89.81 | 1.000 |
| ANN + L2 | - | - | - | - | 87.10 | 87.30 | 87.62 | 1.000 |

distribution. We emphasize that we do not use posterior tempering or any other ad-hoc tricks for any of the Bayesian methods.

### 5.2.3 Results

The results with the fully connected architecture can be found in Tables 2, 3, and 4 and for the convolutional architecture in Tables 5, 6, and 7. Firstly, we see that using the LBBNN-LCRT gives results that are comparable to the baseline LBBNN method, except for FMNIST where it performs a bit worse both with the fully connected and with the convolutional architecture. It is no surprise that these results are similar, as using the LCRT is mainly a computational advantage. Secondly, we note that our LBBNN-FLOW method performs better than the two aforementioned methods in terms of prediction accuracy, on both convolutional and fully connected architectures, while having the most sparse networks. We also see that in terms of accuracy LBBNN-FLOW performs mostly on par with the other two Bayesian architectures, BNN and BNN-FLOW, especially on the fully connected architecture where it gets comparable accuracy even with very sparse networks. In terms of predictive uncertainty handling, in almost all settings the best performance (across all addressed in our design methods) is achieved by one of the LBBNN methods and in

Table 3: Expected calibration error on the fully connected architecture, where we report the minimum, maximum, and median over the ten different runs. The best median results are bold.

| **KMNIST** | Median probability model | | | Dense model | | |
|---|---|---|---|---|---|---|
| Method | min | median | max | min | median | max |
| LBBNN | 0.031 | 0.034 | 0.039 | 0.039 | **0.041** | 0.046 |
| LBBNN-LCRT | 0.072 | 0.076 | 0.079 | 0.118 | 0.120 | 0.123 |
| LBBNN-FLOW | 0.026 | **0.032** | 0.034 | 0.060 | 0.061 | 0.063 |
| BNN-FLOW | - | - | - | 0.098 | 0.102 | 0.106 |
| BNN | - | - | - | 0.069 | 0.072 | 0.075 |
| ANN | - | - | - | 0.077 | 0.081 | 0.092 |
| ANN + L2 | - | - | - | 0.051 | 0.058 | 0.061 |
| **MNIST** | Median probability model | | | Dense model | | |
| Method | min | median | max | min | median | max |
| LBBNN | 0.020 | 0.021 | 0.022 | 0.022 | 0.022 | 0.023 |
| LBBNN-LCRT | 0.029 | 0.030 | 0.032 | 0.047 | 0.049 | 0.051 |
| LBBNN-FLOW | 0.010 | **0.012** | 0.013 | 0.021 | 0.021 | 0.023 |
| BNN-FLOW | - | - | - | 0.033 | 0.035 | 0.036 |
| BNN | - | - | - | 0.027 | 0.028 | 0.029 |
| ANN | - | - | - | 0.016 | **0.017** | 0.018 |
| ANN + L2 | - | - | - | 0.038 | 0.039 | 0.040 |
| **FMNIST** | Median probability model | | | Dense model | | |
| Method | min | median | max | min | median | max |
| LBBNN | 0.026 | 0.028 | 0.032 | 0.023 | **0.026** | 0.031 |
| LBBNN-LCRT | 0.047 | 0.051 | 0.057 | 0.072 | 0.074 | 0.075 |
| LBBNN-FLOW | 0.019 | **0.021** | 0.023 | 0.029 | 0.031 | 0.033 |
| BNN-FLOW | - | - | - | 0.054 | 0.056 | 0.057 |
| BNN | - | - | - | 0.032 | 0.035 | 0.039 |
| ANN | - | - | - | 0.083 | 0.085 | 0.087 |
| ANN + L2 | - | - | - | 0.030 | 0.033 | 0.035 |

the highest share of the cases by LBBNN-FLOW. Also, in most of the cases, the median probability model allows to improve the calibration of the predictions for image datasets. The higher density in general on the convolutional architectures is mainly a result of them being already sparse in the beginning. However, these networks could also be sparsified further by using more conservative priors on inclusions of the weights. We note that the frequentist networks perform slightly worse on these datasets with our chosen architectures. The results could likely be improved by adding more regularization, such as dropout or batch-normalization, but we do not do this here. The increased predictive power of using normalizing flows comes at a computational cost. With the fully connected architecture, we observed that it took around 4 seconds to train one epoch with LBBNN-LCRT, 13 seconds with LBBNN, and 17 seconds with LBBNN-FLOW on an NVIDIA A10 GPU. On the convolutional architecture, it took 7 seconds per epoch with the LBBNN-LCRT, 18 seconds with LBBNN, and 28 with LBBNN-FLOW. Naturally, the frequentist networks are much more computationally efficient, as they only have half the parameters of a standard BNN. These results, however, were observed on a shared computational node and hence could be influenced by loads at different times. Yet they are consistent with theoretical training costs reported in Table 8 in Appendix A.

For the results on the tabular datasets, we refer to Appendix D, where we also report the expected calibration error for classification datasets and pinball loss for regression datasets.

Table 4: Negative test log-likelihood on the fully connected architecture where we report the minimum, maximum, and median over the ten different runs. The best median results are bold.

| **KMNIST** | Median probability model | | | Dense model | | |
|---|---|---|---|---|---|---|
| Method | min | median | max | min | median | max |
| LBBNN | 0.527 | **0.537** | 0.553 | 0.544 | 0.555 | 0.571 |
| LBBNN-LCRT | 0.544 | 0.548 | 0.561 | 0.588 | 0.599 | 0.606 |
| LBBNN-FLOW | 0.569 | 0.589 | 0.626 | 0.605 | 0.621 | 0.670 |
| BNN-FLOW | - | - | - | 0.818 | 0.869 | 0.929 |
| BNN | - | - | - | 0.620 | 0.628 | 0.638 |
| ANN | - | - | - | 1.255 | 1.324 | 1.554 |
| ANN + L2 | - | - | - | 0.417 | **0.424** | 0.434 |
| **MNIST** | Median probability model | | | Dense model | | |
| Method | min | median | max | min | median | max |
| LBBNN | 0.095 | 0.099 | 0.103 | 0.097 | **0.102** | 0.105 |
| LBBNN-LCRT | 0.111 | 0.114 | 0.117 | 0.132 | 0.134 | 0.137 |
| LBBNN-FLOW | 0.091 | **0.094** | 0.104 | 0.103 | 0.109 | 0.114 |
| BNN-FLOW | - | - | - | 0.160 | 0.173 | 0.199 |
| BNN | - | - | - | 0.117 | 0.120 | 0.127 |
| ANN | - | - | - | 0.167 | 0.184 | 0.193 |
| ANN + L2 | - | - | - | 0.119 | 0.120 | 0.122 |
| **FMNIST** | Median probability model | | | Dense model | | |
| Method | min | median | max | min | median | max |
| LBBNN | 0.350 | **0.352** | 0.356 | 0.358 | **0.360** | 0.365 |
| LBBNN-LCRT | 0.422 | 0.425 | 0.435 | 0.439 | 0.441 | 0.443 |
| LBBNN-FLOW | 0.360 | 0.363 | 0.368 | 0.375 | 0.379 | 0.384 |
| BNN-FLOW | - | - | - | 0.416 | 0.421 | 0.423 |
| BNN | - | - | - | 0.515 | 0.519 | 0.530 |
| ANN | - | - | - | 0.794 | 0.850 | 0.864 |
| ANN + L2 | - | - | - | 0.362 | 0.365 | 0.370 |

## 6 Discussion

We have demonstrated that increasing the flexibility in the variational posterior distribution with normalizing flows improves the predictive power compared to the baseline method (with mean-field posterior) while obtaining more sparse networks, despite having a looser variational bound than the mean-field approach. Also, the flow method performed best on a variable selection problem demonstrating better structure learning performance, while the mean-field approaches struggle with highly correlated variables. Also, the calibration of uncertainties in predictive applications is similar with a slight advantage of the proposed approach in this paper. Unlike dense BNNs, our methods have the additional advantage of being able to perform variable selection. The downside is that LBBNNs have an extra parameter per weight, making them less computationally efficient than dense BNNs. Using normalizing flows is a further computational burden as we must also optimize over all the extra flow parameters. If uncertainty handling is not desirable, one could gain the minimal number of predictive parameters using the model trained with flows by relying on the posterior means of the median probability model's parameters. This approach is studied for simpler approximations in more detail in Hubin & Storvik (2024) but it is omitted in this paper. Also, in the future, it would be of interest to develop reversible jump MCMC (Green & Hastie, 2009) or other relevant MCMC methods for LBBNN and compare the variational approximations with the exact sample at least in the low-dimensional cases where the latter could be feasible.

Table 5: Performance metrics on the KMNIST, MNIST, FMNIST validation data, with the convolutional architecture. See the caption in Table 2 for more details.

| KMNIST | Median probability model | | | | Dense model | | | |
|---|---|---|---|---|---|---|---|---|
| Method | min | median | max | density | min | median | max | density |
| LBBNN | 95.39 | 95.59 | 95.75 | 0.359 | 95.35 | 95.56 | 95.65 | 1.000 |
| LBBNN-LCRT | 94.85 | 95.07 | 95.21 | 0.428 | 94.98 | 95.42 | 95.54 | 1.000 |
| LBBNN-FLOW | 95.63 | **95.96** | 96.14 | **0.352** | 95.76 | 95.99 | 96.31 | 1.000 |
| BNN-FLOW | - | - | - | - | 95.89 | **96.38** | 96.52 | 1.000 |
| BNN | - | - | - | - | 95.06 | 95.37 | 95.63 | 1.000 |
| ANN | - | - | - | - | 94.61 | 94.96 | 95.26 | 1.000 |
| ANN + L2 | - | - | - | - | 92.08 | 92.61 | 92.76 | 1.000 |

| MNIST | Median probability model | | | | Dense model | | | |
|---|---|---|---|---|---|---|---|---|
| Method | min | median | max | density | min | median | max | density |
| LBBNN | 99.20 | 99.26 | 99.30 | 0.354 | 99.20 | 99.25 | 99.28 | 1.000 |
| LBBNN-LCRT | 99.05 | 99.25 | 99.29 | 0.406 | 99.23 | 99.30 | 99.36 | 1.000 |
| LBBNN-FLOW | 99.19 | **99.27** | 99.35 | **0.338** | 99.21 | 99.32 | 99.41 | 1.000 |
| BNN-FLOW | - | - | - | - | 99.27 | **99.36** | 99.40 | 1.000 |
| BNN | - | - | - | - | 99.22 | 99.30 | 99.38 | 1.000 |
| ANN | - | - | - | - | 98.99 | 99.13 | 99.18 | 1.000 |
| ANN + L2 | - | - | - | - | 97.95 | 98.29 | 98.42 | 1.000 |

| FMNIST | Median probability model | | | | Dense model | | | |
|---|---|---|---|---|---|---|---|---|
| Method | min | median | max | density | min | median | max | density |
| LBBNN | 90.85 | 91.48 | 91.71 | **0.354** | 90.95 | 91.33 | 91.69 | 1.000 |
| LBBNN-LCRT | 89.94 | 90.39 | 90.79 | 0.433 | 90.53 | 90.82 | 91.34 | 1.000 |
| LBBNN-FLOW | 90.70 | **91.60** | 92.08 | 0.366 | 91.27 | 91.60 | 92.30 | 1.000 |
| BNN-FLOW | - | - | - | - | 91.53 | **91.80** | 92.12 | 1.000 |
| BNN | - | - | - | - | 91.37 | 91.68 | 92.12 | 1.000 |
| ANN | - | - | - | - | 90.49 | 91.07 | 91.73 | 1.000 |
| ANN + L2 | - | - | - | - | 87.97 | 88.08 | 88.46 | 1.000 |

In this paper, we use the same prior for all the weights and inclusion indicators, although this is not necessary. A possible avenue of further research could be to vary the prior inclusion probabilities, to induce different sparsity structures or to incorporate the actual prior knowledge about prior inclusion probabilities of the covariates. Currently, we are taking into account uncertainty in weights and parameters, given some neural network architecture. In the future, it may be of interest to see if it is also possible to incorporate uncertainty in the activation functions. By having skip connections to the output, we could learn with uncertainties to skip all non-linear layers if a linear function is enough, or if a constant estimate of the parameters of the responses is enough (null model), or if one needs some nonlinear layers. This could lead to more transparent Bayesian deep learning models. But the success in that task relies on sufficiently good structure learning, where mean field-approximations are known to not work well (Carbonetto & Stephens, 2012).

A possible application is to do a genome-wide association study (GWAS), using our method. Combining LBBNNs and GWAS has been proposed by Demetci et al. (2021), however, this only uses the mean-field posterior. With our normalizing flow approach, we can easily model dependencies within each SNP set, in addition to dependencies between the different SNP sets. Another set of promising applications is recovering structural equations in nonlinear dynamical systems as the robust and uncertainty-aware alternative to $l1$ penalty used in the Sindy approach (Brunton et al., 2016).

Table 6: Expected calibration error on the convolutional architecture, where we report the minimum, maximum, and median over the ten different runs. The best median results are bold.

| KMNIST | Median probability model | | | Dense model | | |
|---|---|---|---|---|---|---|
| Method | min | median | max | min | median | max |
| LBBNN | 0.041 | 0.045 | 0.048 | 0.050 | 0.051 | 0.053 |
| LBBNN-LCRT | 0.015 | **0.018** | 0.020 | 0.117 | **0.019** | 0.120 |
| LBBNN-FLOW | 0.034 | 0.039 | 0.043 | 0.028 | 0.030 | 0.034 |
| BNN-FLOW | - | - | - | 0.029 | 0.032 | 0.035 |
| BNN | - | - | - | 0.018 | **0.019** | 0.020 |
| ANN | - | - | - | 0.041 | 0.042 | 0.044 |
| ANN + L2 | - | - | - | 0.018 | 0.022 | 0.025 |

| MNIST | Median probability model | | | Dense model | | |
|---|---|---|---|---|---|---|
| Method | min | median | max | min | median | max |
| LBBNN | 0.014 | 0.016 | 0.017 | 0.016 | 0.017 | 0.018 |
| LBBNN-LCRT | 0.005 | **0.005** | 0.006 | 0.006 | 0.006 | 0.008 |
| LBBNN-FLOW | 0.010 | 0.012 | 0.009 | 0.010 | 0.011 | 0.023 |
| BNN-FLOW | - | - | - | 0.009 | 0.009 | 0.010 |
| BNN | - | - | - | 0.005 | **0.005** | 0.006 |
| ANN | - | - | - | 0.007 | 0.008 | 0.009 |
| ANN + L2 | - | - | - | 0.013 | 0.014 | 0.017 |

| FMNIST | Median probability model | | | Dense model | | |
|---|---|---|---|---|---|---|
| Method | min | median | max | min | median | max |
| LBBNN | 0.022 | 0.025 | 0.026 | 0.028 | 0.030 | 0.034 |
| LBBNN-LCRT | 0.013 | **0.018** | 0.021 | 0.018 | **0.020** | 0.021 |
| LBBNN-FLOW | 0.022 | 0.026 | 0.030 | 0.020 | 0.022 | 0.028 |
| BNN-FLOW | - | - | - | 0.023 | 0.027 | 0.031 |
| BNN | - | - | - | 0.021 | 0.022 | 0.025 |
| ANN | - | - | - | 0.059 | 0.064 | 0.068 |
| ANN + L2 | - | - | - | 0.020 | 0.023 | 0.029 |

In our experimental design, we check the effect of direct treatments from Figure 3, especially we were interested in edges from LBBNN to LBBNN-LCRT and to LBBNN-FLOW and from BNN-FLOW to LBNNN-FLOW as the closest neighbors to the proposed in this paper methodology. None of the results is completely uniform across all cases, however we demonstrated that as compared to BNN-FLOW, our proposed approach typically improves on sparsity without losing predictive performance and yields robustly well-calibrated uncertainty handling. Further, as compared to LBBNN, we improve the predictions and sparsity slightly. Improvements are observed in most addressed variable selection tasks when using the LBBNN-FLOW approach as compared to LBBNN-LRCT, while in all variable selection tasks, LBBNN-FLOW with respect to CS approach and in many cases outperforms it, which is especially the case for block diagonal correlation structures in Sections 5.1 and experiment 3 in Appendix B. Methods beyond Figure 3 including other sparsity inducing Bayesian approaches to BNN (Molchanov et al., 2017; Deng et al., 2019; Li et al., 2024), that are less related to our approach and could not be incorporated with direct edges in Figure 3, were omitted in this paper. Yet in the future, following the guidance from Herrmann et al. (2024), it would be interesting to perform a neutral comparison study (Boulesteix et al., 2013). This study should not be done by the authors of any of these papers to avoid biases and tuning one's own methods to beat (in some sense) all competitors and hack the state-of-the-art, which recently became a common practice that often results in poor empirical evidence (Herrmann et al., 2024). This should also be ideally done on new datasets not

Table 7: Negative test log-likelihood on the convolutional architecture where we report the minimum, maximum, and median over the ten different runs. The best median results are bold.

| KMNIST | Median probability model | | | Dense model | | |
|---|---|---|---|---|---|---|
| Method | min | median | max | min | median | max |
| LBBNN | 0.272 | 0.283 | 0.294 | 0.281 | 0.289 | 0.298 |
| LBBNN-LCRT | 0.260 | **0.267** | 0.276 | 0.250 | 0.262 | 0.274 |
| LBBNN-FLOW | 0.324 | 0.352 | 0.380 | 0.280 | 0.308 | 0.331 |
| BNN-FLOW | - | - | - | 0.289 | 0.314 | 0.354 |
| BNN | - | - | - | 0.306 | 0.339 | 0.360 |
| ANN | - | - | - | 0.467 | 0.525 | 0.576 |
| ANN + L2 | - | - | - | 0.250 | **0.259** | 0.277 |
| **MNIST** | Median probability model | | | Dense model | | |
| Method | min | median | max | min | median | max |
| LBBNN | 0.043 | 0.045 | 0.048 | 0.045 | 0.047 | 0.050 |
| LBBNN-LCRT | 0.029 | **0.030** | 0.034 | 0.029 | **0.030** | 0.034 |
| LBBNN-FLOW | 0.046 | 0.049 | 0.054 | 0.038 | 0.041 | 0.045 |
| BNN-FLOW | - | - | - | 0.040 | 0.042 | 0.043 |
| BNN | - | - | - | 0.033 | 0.036 | 0.039 |
| ANN | - | - | - | 0.046 | 0.050 | 0.060 |
| ANN + L2 | - | - | - | 0.057 | 0.060 | 0.067 |
| **FMNIST** | Median probability model | | | Dense model | | |
| Method | min | median | max | min | median | max |
| LBBNN | 0.276 | **0.282** | 0.291 | 0.284 | 0.293 | 0.300 |
| LBBNN-LCRT | 0.284 | 0.298 | 0.315 | 0.271 | 0.283 | 0.301 |
| LBBNN-FLOW | 0.345 | 0.360 | 0.374 | 0.310 | 0.324 | 0.342 |
| BNN-FLOW | - | - | - | 0.367 | 0.402 | 0.459 |
| BNN | - | - | - | 0.270 | **0.277** | 0.288 |
| ANN | - | - | - | 0.470 | 0.507 | 0.557 |
| ANN + L2 | - | - | - | 0.332 | 0.337 | 0.341 |

used in the papers describing the compared methods. For a non-neutral comparison of LBBNN with other sparse neural networks, a curious reader can check the results in Hubin & Storvik (2019; 2024).

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

## Supplementary material

The code used for the experiments can be found in the accompanying zip folder and on GitHub `https://github.com/LarsELund/Sparsifying-BNNs-with-LRT-and-NF`.

# A    Computational and memory efficiency

Table 8: Training Cost (per iteration of first order optimizer) and Memory Cost (for saving the trained model's parameters) for Different Methods. Assume (only) for this table the following notation: $n$ - number of data points in the minibatch; $p$ - number of trainable parameters in the architecture; $s$ - number of samples from parameters for gradient estimation; $f$ - additional flow parameters; $w$ - speed-up factor due to sampling directly from neurons (number of weights per neuron); $\alpha$ - ratio of non-pruned weights. The training iteration cost reflects the computational complexity for each method. The memory cost includes considerations for additional parameters and storage efficiency, especially for pruned weights. A non-pruned weight is assumed to need a double-precision number to be stored (64 bits). In the pruned model, we further assume to need 1 bit per position of whether the weight there is pruned.

| Method | Training Cost | Memory Cost (bits) | Sparse Memory Cost (bits) |
|---|---|---|---|
| **ANN** | $\mathcal{O}(n \cdot p)$ | $64 \cdot p$ | Not applicable |
| **BNN** | $\mathcal{O}(n \cdot 2p \cdot s)$ | $64 \cdot 2p$ | Not applicable |
| **BNN-FLOW** | $\mathcal{O}(n \cdot (2p + f) \cdot s/w)$ | $64 \cdot (2p + f)$ | Not applicable |
| **LBBNN** | $\mathcal{O}(n \cdot 3p \cdot s)$ | $64 \cdot 3p$ | $\alpha \cdot 2p + 65 \cdot (1 - \alpha) \cdot 2p$ |
| **LBBNN-LRCT** | $\mathcal{O}(n \cdot 3p \cdot s/w)$ | $64 \cdot 3p$ | $\alpha \cdot 2p + 65 \cdot (1 - \alpha) \cdot 2p$ |
| **LBBNN-FLOW** | $\mathcal{O}(n \cdot (3p + f) \cdot s/w)$ | $64 \cdot (2p + f)$ | $\alpha \cdot 2p + 65 \cdot ((1 - \alpha) \cdot 2p) + 64 \cdot f$ |

# B    Additional simulation studies

In this section, we describe three additional simulation studies, conducted to evaluate the performance of Bayesian variable selection using the same approaches as in Section 5.1 of the paper. Just like in the extension from Section 5.1.1., the studies involved generating simulated datasets and assessing the True Positive Rate (TPR) and False Positive Rate (FPR) of selected by different methods variables across 3 different signal-to-noise ratios and 10 correlation structures per signal level. 20 simulations on different seeds were run per setting per method, resulting in 600 simulations per method in each of these three studies. Bayesian variable selection was performed using the median probability model. We do not tune any methods but rather use exactly the same hyperparameters for each experiment as we did in Section 5.1. However, we use different data-generative processes in each study. Below, we explain these processes.

## B.1    Data generating process

### B.1.1    Experiment 1:

For the first experiment, we use the Simrel package (Sæbø et al., 2015), which allows for simulating correlated data where the level of correlation can be controlled through the $\gamma$ parameter, which is the declining (decaying) factor of eigenvalues of predictors. The parameter was ranging between 0.1 and 1. For this experiment, we have 50 predictor variables, and sample 1000 data points, with the true effect vector is defined as:

$$\boldsymbol{\beta} = \left( \underbrace{1, 1, \ldots, 1}_{10 \text{ true signals}}, \underbrace{0, 0, \ldots, 0}_{40 \text{ noise covariates}} \right). \tag{13}$$

We then generate the continuous outcome as: $\eta_i = \boldsymbol{\beta}^\top \mathbf{X}_i + \epsilon_i, \quad \epsilon_i \sim \mathcal{N}(0, \sigma^2), \sigma \in \{10, 1, 0.1\}$. Finally, we convert the continuous outcome $\eta$ to a binary outcomes as $y_i = \mathbb{I}(\eta_i > \text{median}(\eta))$.

### B.1.2    Experiment 2:

Here, we simulate i.i.d. $x_{i,j} \sim \mathcal{N}(0, 1)$, then induce the following correlation structure: For a correlation parameter $\phi \in \{1, 0.9, \ldots, 0.1\}$, update the covariates as: $x_{i,j} = \rho \cdot x_{i,j-10} + (1 - \rho) \cdot x_{i,j}, i = 1, \ldots, 1000, j = 11, \ldots, 50$. The resulting predictors are then standardized, followed by binarizing all even indexed covariates

using the median threshold, i.e. $x_{i,j} = \mathbb{I}(x_{i,j} > \text{median}(\mathbf{x_j})), j \in \{2,4,...,50\}$. We further define the true effect vector $\boldsymbol{\beta}$ as:

$$\boldsymbol{\beta} = \left( \underbrace{\beta_1, \ldots, \beta_{10}}_{\text{10 true signals}}, \underbrace{0,0,\ldots,0}_{\text{40 noise covariates}} \right), \tag{14}$$

with $\beta_1, \ldots \beta_{10} \sim \mathcal{N}(1,1)$. We then generate the binary outcome the same way as in Experiment 1, but with $\sigma \in \{50, 10, 1\}$.

### B.1.3 Experiment 3:

Finally, we take inspiration from the Rejoinder to Hubin et al. (2020) and simulate Quantitative trait locus (QTL) mapping using R/QTL package (Broman et al., 2003). Here, just as in Section 5.1, we have a block-diagonal correlation structure between the covariates. Further, in this experiment, we randomly select each $\beta_i$ to be a true parameter with probability of 0.25. The true $\beta_i$ are then sampled as $\beta_i \sim \mathcal{N}(1,1)$. Here, we generate the response as: $\eta_i = \mathcal{N}(\boldsymbol{\beta}^\top \mathbf{X}_i, \sigma_i^2)$, $y_i \sim \text{Bernoulli}\left(\frac{1}{1+\exp(-\eta_i)}\right)$, and we use $\sigma \in \{1, 0.1, 0.01\}$.

### B.2 Results

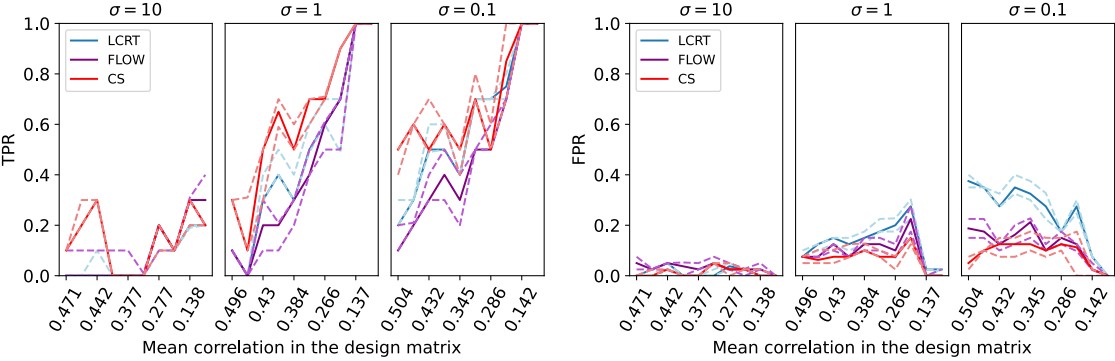

Figure 7: On the left, true positive rates, with false positive rates to the right. The maximum absolute pairwise correlations between the covariates ranged between 0.102 and 0.994.

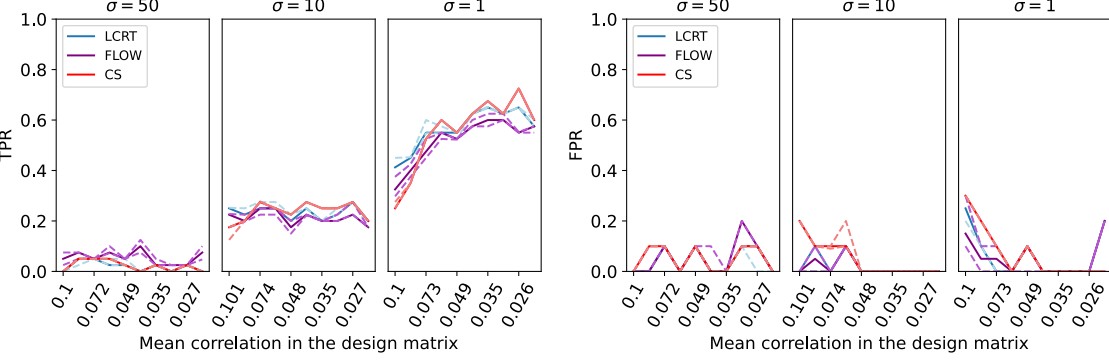

Figure 8: On the left, true positive rates, with false positive rates to the right. The maximum absolute pairwise correlations between the covariates ranged between 0.102 and 0.995.

The results for these additional simulation experiments 1,2 and 3 can be found in Figures 7, 8, and 9 respectively. In experiment 1, we see that the CS method generally has a higher TPR, whereas the results

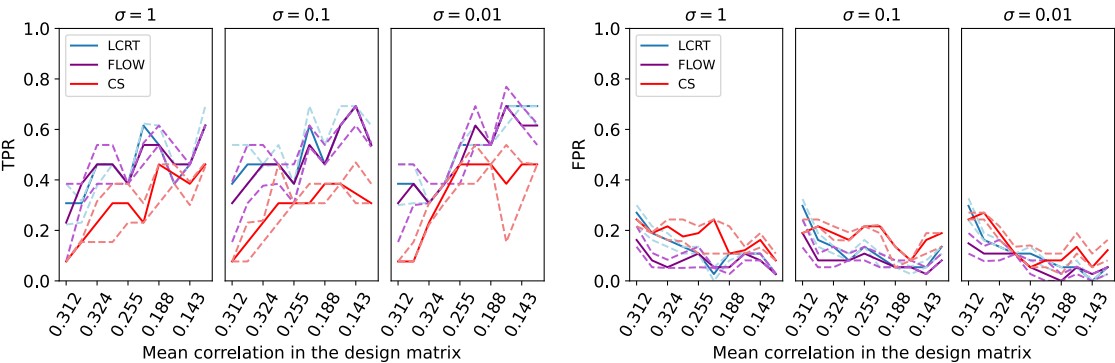

Figure 9: On the left, true positive rates, with false positive rates to the right. The maximum absolute pairwise correlations between the covariates ranged between 0.909 and 0.999.

are closer on the FPRs, with the LCRT being slightly worse. In the second experiment, the results are quite close, with flows having a slightly higher TPR for high $\sigma$ levels, and slightly lower with smaller $\sigma$. In the last experiment, we see that FLOW generally has the lowest FPR and highest TPR. As indicated in experiment 3 and also in Sections 5.1 and 5.1.1, we can see the advantage of using flows when the covariance structure between the covariates is block-diagonal.

## C   Convolutional architectures

For convolutional layers, the variational distribution is defined to be:

$$q_{\boldsymbol{\theta}}(\mathbf{W}|\boldsymbol{\Gamma}, \boldsymbol{z}) = \prod_{i=1}^{n_h} \prod_{j=1}^{n_w} \prod_{k=1}^{n_f} [\gamma_{ijk} \mathcal{N}(w_{ijk}; z_k \tilde{\mu}_{ijk}, \tilde{\sigma}_{ijk}^2) + (1 - \gamma_{ijk})\delta(w_{ijk})]$$

$$q_{\tilde{\alpha}_{ijk}}(\gamma_{ijk}) = \text{Bernoulli}(\gamma_{ijk}; \tilde{\alpha}_{ijk}),$$

(15)

where $n_h$, $n_w$, and $n_f$ denote the height, width, and number of filters in the convolutional kernel.

For the convolutional layers, we use the following for the inverse normalizing flows:

$$\boldsymbol{\nu} = ((\text{Mat}(\boldsymbol{W} \odot \boldsymbol{\Gamma})\mathbf{e}) \otimes \mathbf{d}_1)\,(\mathbf{1} \odot (n_h n_w)^{-1})$$

$$\log \boldsymbol{\tau^2} = ((\text{Mat}(\boldsymbol{W} \odot \boldsymbol{\Gamma})\mathbf{e}) \otimes \mathbf{d}_2)\,(\mathbf{1} \odot (n_h n_w)^{-1}).$$

(16)

Here, Mat($\cdot$) denotes the matricisation operator (as defined in Louizos & Welling (2017)), i.e. changing the shape of a multidimensional tensor into a matrix.

## D   Experiments on tabular datasets

In this experiment, we consider six classification datasets and three regression datasets. We compare our approach LBBNN-FLOW against LBBNN-LCRT, LBBNN, a dense BNN, BNN-FLOW, ANN+L2, ANN, and Monte Carlo (MC) dropout (with dropout rates corresponding to our prior inclusion probabilities), in addition to Gaussian processes for the regression datasets. Here, we use an out-of-the-box version of the package from Varvia et al. (2023), using the Matérn 3/2 covariance function (see chapter 4 of Rasmussen (2003) for how this is defined). For the mean function, we use the training data average. For the covariance function, the implementation defines three hyperparameters: the kernel variance and length scale, controlling the smoothness properties of the regression, and the error variance parameter, regulating how closely to fit the training data. We define these parameters to be 1, 5, and 0.1 respectively.

For the neural networks, we use a single hidden layer with 500 neurons and again train for 250 epochs with the Adam optimizer. We use 10-fold cross-validation and report the minimum, mean and maximum accuracy

over these 10 repetitions, in addition to the mean sparsity. We also report the expected calibration error (ECE), in addition to $p_{\mathrm{WAIC}_1}$ and $p_{\mathrm{WAIC}_2}$ for the classification datasets, while for the regression datasets we report RMSE, the pinball loss, $p_{\mathrm{WAIC}_1}$ and $p_{\mathrm{WAIC}_2}$.

For the six classification datasets, most are taken from the UCI machine learning repository. The Credit Approval dataset (Quinlan) consists of 690 samples with 15 variables, with the response variable being whether someone gets approved for a credit card or not. The Bank Marketing dataset (Moro et al., 2012) consists of data (45211 samples and 17 variables) related to a marketing campaign of a Portuguese banking institution, where the goal is to classify whether the persons subscribed to the service or not. In addition to this, we use the Census Income dataset (Kohavi, 1996) with 48842 samples and 14 variables, where we try to classify whether someone's income exceeds 50000 dollars per year. Additionally, we have three datasets related to classifying food items. The first, the Raisins dataset (Çinar et al., 2020), consists of 900 samples and 7 variables, where the goal is to classify into two different types of raisins grown in Turkey. Secondly, we use the Dry Beans dataset (UCI, 2020), consisting of 13611 samples, 17 variables, and 7 different types of beans. Lastly, the Pistachio dataset (Ozkan et al., 2021) consists of 2148 samples and 28 variables, with two different types of Pistachios.

For the regression datasets, we use the abalone shell data (Nash et al., 1995), the wine quality dataset (Cortez et al., 2009), and the Boston housing data (Harrison Jr & Rubinfeld, 1978). To avoid model misspecification, responses were standardized for the regression datasets. The variance of the responses was then assumed fixed and equal to 1 in all of the models, while the mean parameter was modeled.

The predictive accuracy and density for the classification datasets can be found in Table 9 and Table 10, and the expected calibration error in Table 11. Finally, we have the $p_{\mathrm{WAIC}_1}$ and $p_{\mathrm{WAIC}_2}$ metrics in Table 12 and 13, respectively. For the regression datasets, we have the RMSE and density results in Table 14, in addition to pinball loss in Table 15 and $p_{\mathrm{WAIC}_1}$ and $p_{\mathrm{WAIC}_2}$ in Table 16.

For the classification datasets, we see that our LBBNN-FLOW method typically performs well compared to the LBBNN baseline, mostly having higher predictive power, also with the median probability model. It also performs well compared to our other baseline methods. On calibration, we see that our method again performs well compared to baselines. Finally, for the $p_{\mathrm{WAIC}_1}$ and $p_{\mathrm{WAIC}_2}$, we see that using the median probability model typically reduces this metric. We note that our method often has lower values than BNN-FLOW and BNN, even though these methods have fewer parameters. As mentioned before, however, it is unclear how good of an estimate this is for the effective number of parameters in highly non-linear methods such as neural networks. For the regression examples, our method performs slightly worse than some baselines in terms of RMSE and pinball loss, however, LBBNN-LCRT performs best (on average) on one dataset.

Table 9: Performance results on the Credit Approval, Bank Marketing, and Census Income datasets, using 10-fold cross-validation. The minimum, mean and maximum accuracies are reported, in addition to the density. The best results are bold.

| **Credit Approval** | Median probability model | | | | Dense model | | | |
|---|---|---|---|---|---|---|---|---|
| Method | min | mean | max | density | min | mean | max | density |
| LBBNN | 81.16 | 85.22 | 91.30 | 0.431 | 81.16 | 85.80 | 91.30 | 1.000 |
| LBBNN-LCRT | 81.16 | 86.23 | 92.75 | 0.347 | 81.16 | 86.23 | 91.30 | 1.000 |
| LBBNN-FLOW | 84.10 | **88.55** | 94.20 | 0.348 | 82.61 | **87.68** | 91.30 | 1.000 |
| BNN-FLOW | - | - | - | - | 82.61 | 86.23 | 89.86 | 1.000 |
| BNN | - | - | - | - | 78.26 | 83.33 | 88.41 | 1.000 |
| ANN | - | - | - | - | 78.26 | 83.19 | 91.30 | 1.000 |
| ANN + L2 | - | - | - | - | 73.91 | 83.19 | 89.86 | 1.000 |
| ANN + MC dropout | - | - | - | - | 81.16 | 86.52 | 92.75 | 1.000 |

| **Bank Marketing** | Median probability model | | | | Dense model | | | |
|---|---|---|---|---|---|---|---|---|
| Method | min | mean | max | density | min | mean | max | density |
| LBBNN | 89.75 | 90.66 | 91.74 | 0.430 | 89.75 | 90.61 | 91.43 | 1.000 |
| LBBNN-LCRT | 90.75 | 91.27 | 92.16 | 0.347 | 90.60 | 91.27 | 92.16 | 1.000 |
| LBBNN-FLOW | 90.58 | **91.38** | 92.08 | 0.347 | 90.75 | **91.36** | 92.03 | 1.000 |
| BNN-FLOW | - | - | - | - | 90.14 | 91.13 | 91.96 | 1.000 |
| BNN | - | - | - | - | 90.63 | 91.16 | 91.74 | 1.000 |
| ANN | - | - | - | - | 90.93 | 90.97 | 91.62 | 1.000 |
| ANN + L2 | - | - | - | - | 90.75 | 91.15 | 91.77 | 1.000 |
| ANN + MC dropout | - | - | - | - | 90.70 | 91.17 | 92.06 | 1.000 |

| **Cencus Income** | Median probability model | | | | Dense model | | | |
|---|---|---|---|---|---|---|---|---|
| Method | min | mean | max | density | min | mean | max | density |
| LBBNN | 85.24 | 85.74 | 86.49 | 0.431 | 85.52 | 85.85 | 86.69 | 1.000 |
| LBBNN-LCRT | 85.36 | 85.90 | 86.77 | 0.349 | 85.63 | 85.92 | 86.57 | 1.000 |
| LBBNN-FLOW | 85.60 | **86.04** | 86.43 | 0.349 | 85.57 | **86.05** | 86.49 | 1.000 |
| BNN-FLOW | - | - | - | - | 85.05 | 85.32 | 85.79 | 1.000 |
| BNN | - | - | - | - | 84.77 | 85.27 | 86.45 | 1.000 |
| ANN | - | - | - | - | 84.56 | 85.09 | 85.54 | 1.000 |
| ANN + L2 | - | - | - | - | 84.73 | 85.39 | 85.91 | 1.000 |
| ANN + MC dropout | - | - | - | - | 85.32 | 85.89 | 86.49 | 1.000 |

Table 10: Performance results on the Dry Beans, Pistachio, and Raisin datasets, using 10-fold cross-validation. The minimum, mean and maximum accuracies are reported, in addition to the density. The best results are bold.

| **Dry Beans** | Median probability model | | | | Dense model | | | |
|---|---|---|---|---|---|---|---|---|
| Method | min | mean | max | density | min | mean | max | density |
| LBBNN | 90.88 | 92.65 | 93.90 | 0.442 | 91.25 | 92.80 | 93.82 | 1.000 |
| LBBNN-LCRT | 91.62 | **93.18** | 94.41 | 0.349 | 91.69 | 93.34 | 94.34 | 1.000 |
| LBBNN-FLOW | 89.26 | 92.38 | 93.90 | 0.279 | 89.85 | 92.57 | 94.19 | 1.000 |
| BNN-FLOW | - | - | - | - | 91.32 | 93.05 | 94.19 | 1.000 |
| BNN | - | - | - | - | 91.47 | 93.35 | 94.63 | 1.000 |
| ANN | - | - | - | - | 91.47 | 93.37 | 94.71 | 1.000 |
| ANN + L2 | - | - | - | - | 91.54 | **93.38** | 94.71 | 1.000 |
| ANN + MC dropout | - | - | - | - | 91.40 | 92.96 | 94.04 | 1.000 |

| **Pistachio** | Median probability model | | | | Dense model | | | |
|---|---|---|---|---|---|---|---|---|
| Method | min | mean | max | density | min | mean | max | density |
| LBBNN | 91.12 | 93.46 | 96.26 | 0.433 | 91.12 | 93.36 | 95.33 | 1.000 |
| LBBNN-LCRT | 91.59 | **93.93** | 95.79 | 0.350 | 92.06 | 94.07 | 95.79 | 1.000 |
| LBBNN-FLOW | 91.12 | 93.46 | 95.33 | 0.350 | 91.12 | 93.46 | 95.33 | 1.000 |
| BNN-FLOW | - | - | - | - | 90.65 | 93.60 | 96.26 | 1.000 |
| BNN | - | - | - | - | 92.52 | 94.07 | 96.26 | 1.000 |
| ANN | - | - | - | - | 92.06 | 94.11 | 96.73 | 1.000 |
| ANN + L2 | - | - | - | - | 92.06 | 93.93 | 96.26 | 1.000 |
| ANN + MC dropout | - | - | - | - | 91.12 | **94.16** | 96.26 | 1.000 |

| **Raisins** | Median probability model | | | | Dense model | | | |
|---|---|---|---|---|---|---|---|---|
| Method | min | mean | max | density | min | mean | max | density |
| LBBNN | 83.33 | **87.00** | 92.22 | 0.439 | 83.33 | 86.78 | 92.22 | 1.000 |
| LBBNN-LCRT | 81.11 | 86.11 | 91.11 | 0.349 | 81.11 | 86.78 | 92.22 | 1.000 |
| LBBNN-FLOW | 83.33 | 86.67 | 92.22 | 0.349 | 82.22 | 86.56 | 92.22 | 1.000 |
| BNN-FLOW | - | - | - | - | 82.22 | 87.22 | 91.11 | 1.000 |
| BNN | - | - | - | - | 81.11 | **87.89** | 92.22 | 1.000 |
| ANN | - | - | - | - | 81.11 | 86.44 | 90.00 | 1.000 |
| ANN + L2 | - | - | - | - | 81.11 | 87.56 | 92.22 | 1.000 |
| ANN + MC dropout | - | - | - | - | 81.11 | 87.00 | 93.33 | 1.000 |

Table 11: Expected calibration error, with minimum, mean, and maximum values obtained using 10-fold cross-validation. MPM denotes the medium probability model.

| Method | ECE (min, mean, max) | | | | | |
|---|---|---|---|---|---|---|
| | Credit Approval | Bank Marketing | Census Income | Dry Beans | Pistachio | Raisins |
| LBBNN | (0.056, 0.079, 0.127) | (0.023, 0.031, 0.039) | (0.005, **0.010**, 0.015) | (0.011, 0.016, 0.021) | (0.012, 0.028, 0.047) | (0.043, 0.073, 0.097) |
| LBBNN-MPM | (0.035, 0.080, 0.122) | (0.013, 0.025, 0.033) | (0.006, 0.011, 0.016) | (0.002, 0.015, 0.031 ) | (0.014, 0.028, 0.043) | (0.035, 0.069, 0.103) |
| LBBNN-LCRT | (0.035, 0.071,0.097) | (0.001, 0.016, 0.020) | (0.006, **0.010**, 0.012) | (0.008, 0.014, 0.019) | (0.014, **0.022**, 0.034) | (0.034, 0.070, 0.094) |
| LBBNN-LCRT-MPM | (0.015, 0.073, 0.119) | (0.011, 0.015, 0.018) | (0.008, 0.012, 0.016) | (0.007, **0.013**, 0.025) | (0.012, 0.025, 0.035) | (0.057, 0.076, 0.115) |
| LBBNN-FLOW | (0.030, **0.069**, 0.103) | (0.002, **0.007**, 0.015) | (0.007, 0.011, 0.016) | (0.010, 0.017, 0.040) | (0.011, 0.024, 0.045) | (0.018, 0.068, 0.103) |
| LBBNN-FLOW-MPM | (0.025, 0.071, 0.126) | (0.003, 0.008, 0.012) | (0.009, 0.012, 0.015 ) | (0.008, 0.014, 0.026) | (0.012, 0.030, 0.047) | (0.038, 0.072, 0.096) |
| BNN-FLOW | (0.025, 0.074, 0.123) | (0.006, 0.012, 0.022) | (0.020, 0.027, 0.047) | (0.007, 0.014, 0.029) | (0.010, 0.029, 0.055) | (0.038, 0.069, 0.104) |
| BNN | (0.046, 0.097, 0.195) | (0.010, 0.015, 0.024) | (0.020, 0.025, 0.031) | (0.005, **0.013**, 0.030) | (0.010, 0.039, 0.057) | (0.034, 0.070, 0.103) |
| ANN | (0.072, 0.131, 0.188) | (0.012, 0.017, 0.023) | (0.020, 0.025, 0.031) | (0.007, **0.013**, 0.021) | (0.027, 0.042, 0.056) | (0.040, 0.074, 0.109) |
| ANN + L2 | (0.020, 0.114, 0.178) | (0.013, 0.018, 0.024) | (0.015, 0.020, 0.026) | (0.007, **0.013**, 0.020) | (0.014, 0.040, 0.053) | (0.025, **0.066**, 0.111) |
| ANN + MC dropout | (0.028, 0.079, 0.115) | (0.011, 0.014, 0.017) | (0.008, 0.011, 0.018) | (0.033, 0.039, 0.048) | (0.016, 0.029, 0.051) | (0.027, 0.069, 0.113) |

Table 12: The $p_{\mathrm{WAIC}_1}$ metric, on our six classification datasets, where the minimum, mean, and maximum values are obtained with 10-fold cross-validation.

| | $p_{\mathrm{WAIC}_1}$ (min, mean, max) | | | | | |
|---|---|---|---|---|---|---|
| Method | Credit Approval | Bank Marketing | Census Income | Dry Beans | Pistachio | Raisins |
| LBBNN | (0.620, 1.323, 2.595) | (28.18, 31.15, 34,74) | (22.98, 29.33, 37.14) | (15.82, 23.18, 30.44) | (2.594, 3.277, 4.762) | (0.149, 0.271, 0.668) |
| **LBBNN-MPM** | **(0.000, 0.000, 0.000)** | **(0.001, 0.002, 0.003)** | **(0.002, 0.002, 0.003)** | **(0.001, 0.002, 0.003)** | **(0.000, 0.000, 0.000)** | **(0.000, 0.000, 0.000)** |
| LBBNN-LCRT | (0.013, 0.026, 0.041) | (0.575, 0.643, 0.725) | (0.997, 1.144, 1.357) | (9.937, 13.32, 20.12) | (0.015, 0.034, 0.052) | (0.011, 0.023, 0.043) |
| LBBNN-LCRT-MPM | (0.011, 0.025, 0.043) | (0.566, 0.645, 0.718) | (1.013, 1.140, 1.180) | (0.002, 0.002, 0.003) | (0.016, 0.032, 0.054) | (0.011, 0.024, 0.045) |
| LBBNN-FLOW | (0.009, 0.021, 0.033) | (0.486, 0.542, 0.608) | (1.045, 1.216, 1.488) | (12.83, 17.66, 26.65) | (0.016, 0.034, 0.058) | (0.011, 0.025, 0.052) |
| LBBNN-FLOW-MPM | (0.009, 0.022, 0.037) | (0.476, 0.544, 0.621) | (1.065, 1.245, 1.501) | (0.331, 0.420, 0.567) | (0.013, 0.033, 0.055) | (0.011, 0.026, 0.055) |
| BNN-FLOW | (0.012, 0.029, 0.054) | (0.523, 0.593, 0.655) | (1.235, 1.398, 1.745) | (0.402, 0.599, 1.089) | (0.017, 0.039, 0.062) | (0.012, 0.023, 0.045) |
| BNN | (0.015, 0.055, 0.255) | (0.580, 0.686, 0.760) | (1.165, 1.352, 1.442) | (0.005, 0.007, 0.009) | (0.019, 0.048, 0.073) | (0.011, 0.022, 0.034) |
| MC dropout | (0.016, 0.058, 0.249) | (0.536, 0.600, 0.687) | (1.057, 1.201, 1.353) | (75.99, 105.3, 133.2) | (0.027, 0.040, 0.059) | (0.011, 0.027, 0.042) |

Table 13: The $p_{\mathrm{WAIC}_2}$ metric, on our six classification datasets, where the minimum, mean, and maximum values are obtained with 10-fold cross-validation.

| | $p_{\mathrm{WAIC}_2}$ (min, mean, max) | | | | | |
|---|---|---|---|---|---|---|
| Method | Credit Approval | Bank Marketing | Census Income | Dry Beans | Pistachio | Raisins |
| LBBNN | (0.649, 1.367, 2.711) | (28.18, 31.73, 35,74) | (23.20, 65.55, 396.7) | (16.69, 24.55, 32.75) | (2.724, 3.485, 5.058) | (0.150, 0.269, 0.647) |
| **LBBNN-MPM** | **(0.000, 0.000, 0.000)** | **(0.001, 0.002, 0.003)** | **(0.002, 0.002, 0.003)** | **(0.001 0.002, 0.003)** | **(0.000, 0.000, 0.000)** | **(0.000, 0.000, 0.000)** |
| LBBNN-LCRT | (0.021, 0.058, 0.111) | (1.016, 1.176, 1.437) | (1.588, 2.981, 12.04) | (10.38, 13.86, 20.86) | (0.023, 0.078, 0.138) | (0.015, 0.044, 0.101) |
| LBBNN-LCRT-MPM | (0.017, 0.056, 0.114) | (0.985, 1.188, 1.374) | (1.1639, 2.041, 2.247) | (0.002, 0.002, 0.003) | (0.026, 0.073, 0.151) | (0.016, 0.047, 0.112) |
| LBBNN-FLOW | (0.014, 0.043, 0.072) | (0.780, 0.926, 1.154) | (1.718, 7.906, 21.91) | (13.30, 18.88, 29.37) | (0.025, 0.071, 0.137) | (0.016, 0.058, 0.197) |
| LBBNN-FLOW-MPM | (0.014, 0.047, 0.083) | (0.755, 0.934, 1.193) | (1.781, 8.940, 21.97) | (0.331, 0.421, 0.569) | (0.018, 0.070, 0.126) | (0.016, 0.059, 0.188) |
| BNN-FLOW | (0.020, 0.074, 0.199) | (0.856, 1.049, 1.303) | (2.230, 4.745, 12.83) | (0.403, 0.601, 1.093) | (0.030, 0.097, 0.178) | (0.016, 0.046, 0.123) |
| BNN | (0.024, 1.087, 10.14) | (1.036, 1.307, 1.517) | (2.052, 3.591, 12.34) | (0.005, 0.007, 0.009) | (0.034, 0.137, 0.244) | (0.015, 0.039, 0.064) |
| MC dropout | (0.027, 1.112, 10.13) | (0.905, 1.059, 1.351) | (1.767, 4.148, 12.03) | (101.0, 148.4, 265.8) | (0.054, 0.102, 0.216) | (0.015, 0.064, 0.181) |

Table 14: Root mean squared error, using 10-fold cross-validation. The minimum, mean and maximum are reported, in addition to the density. The best results are bold.

| **Abalone** | Median probability model | | | | Dense model | | | |
|---|---|---|---|---|---|---|---|---|
| Method | min | mean | max | density | min | mean | max | density |
| LBBNN | 0.597 | 0.677 | 0.799 | 0.435 | 0.577 | 0.665 | 0.795 | 1.000 |
| LBBNN-LCRT | 0.564 | **0.644** | 0.736 | 0.350 | 0.560 | **0.641** | 0.722 | 1.000 |
| LBBNN-FLOW | 0.577 | 0.660 | 0.767 | 0.350 | 0.572 | 0.657 | 0.761 | 1.000 |
| BNN-FLOW | - | - | - | - | 0.585 | 0.654 | 0.733 | 1.000 |
| BNN | - | - | - | - | 0.579 | 0.651 | 0.759 | 1.000 |
| ANN | - | - | - | - | 0.575 | 0.657 | 0.801 | 1.000 |
| ANN + L2 | - | - | - | - | 0.572 | 0.652 | 0.764 | 1.000 |
| ANN + MC dropout | - | - | - | - | 0.579 | 0.655 | 0.759 | 1.000 |
| Gaussian process | - | - | - | - | 0.570 | 0.650 | 0.734 | 1.000 |

| **Wine Quality** | Median probability model | | | | Dense model | | | |
|---|---|---|---|---|---|---|---|---|
| Method | min | mean | max | density | min | mean | max | density |
| LBBNN | 0.768 | 0.805 | 0.854 | 0.435 | 0.757 | 0.799 | 0.845 | 1.000 |
| LBBNN-LCRT | 0.742 | **0.782** | 0.820 | 0.350 | 0.741 | 0.780 | 0.820 | 1.000 |
| LBBNN-FLOW | 0.751 | 0.788 | 0.822 | 0.351 | 0.750 | 0.787 | 0.823 | 1.000 |
| BNN-FLOW | - | - | - | - | 0.747 | 0.778 | 0.815 | 1.000 |
| BNN | - | - | - | - | 0.749 | 0.767 | 0.790 | 1.000 |
| ANN | - | - | - | - | 0.740 | 0.761 | 0.792 | 1.000 |
| ANN + L2 | - | - | - | - | 0.746 | 0.763 | 0.789 | 1.000 |
| ANN + MC dropout | - | - | - | - | 0.758 | 0.799 | 0.847 | 1.000 |
| Gaussian process | - | - | - | - | 0.700 | **0.739** | 0.777 | 1.000 |

| **Boston Housing** | Median probability model | | | | Dense model | | | |
|---|---|---|---|---|---|---|---|---|
| Method | min | mean | max | density | min | mean | max | density |
| LBBNN | 0.267 | 0.412 | 0.621 | 0.431 | 0.262 | 0.393 | 0.552 | 1.000 |
| LBBNN-LCRT | 0.245 | **0.374** | 0.560 | 0.350 | 0.233 | 0.366 | 0.534 | 1.000 |
| LBBNN-FLOW | 0.267 | 0.402 | 0.595 | 0.352 | 0.253 | 0.395 | 0.561 | 1.000 |
| BNN-FLOW | - | - | - | - | 0.241 | 0.363 | 0.524 | 1.000 |
| BNN | - | - | - | - | 0.240 | 0.351 | 0.481 | 1.000 |
| ANN | - | - | - | - | 0.247 | 0.364 | 0.597 | 1.000 |
| ANN + L2 | - | - | - | - | 0.234 | **0.346** | 0.487 | 1.000 |
| ANN + MC dropout | - | - | - | - | 0.244 | 0.371 | 0.502 | 1.000 |
| Gaussian process | - | - | - | - | 0.217 | 0.349 | 0.444 | 1.000 |

Table 15: Mean pinball loss on a grid between 0.05 and 0.95 in increments of 0.05. Min, mean, and max values were obtained using 10-fold cross-validation.

| | Mean pinball (min, mean, max) | | |
|---|---|---|---|
| Method | Abalone | Wine Quality | Boston Housing |
| LBBNN | (0.210, 0.236, 0.260) | (0.291, 0.310, 0.325) | (0.100, 0.133, 0.164) |
| LBBNN-MPM | (0.220, 0.242, 0.260) | (0.296, 0.313, 0.328) | (0.097, 0.137, 0.167) |
| LBBNN-LCRT | (0.203, **0.227**, 0.251) | (0.289, 0.304, 0.316) | (0.087, 0.121, 0.150) |
| LBBNN-LCRT-MPM | (0.204, 0.228, 0.259) | (0.290, 0.304, 0.318) | (0.089, 0.123, 0.151) |
| LBBNN-FLOW | (0.208, 0.232, 0.253) | (0.289 0.306 0.321) | (0.091, 0.132, 0.158) |
| LBBNN-FLOW-MPM | (0.209, 0.231, 0.261) | (0.290, 0.306 0.319) | (0.092, 0.133, 0.154) |
| BNN-FLOW | (0.211, 0.232, 0.250) | (0.293, 0.303, 0.317) | (0.085, 0.120, 0.142) |
| BNN | (0.210, 0.230, 0.255) | (0.286, 0.295, 0.307) | (0.084, 0.117, 0.145) |
| ANN | (0.210, 0.231, 0.255) | (0.279, 0.292, 0.301) | (0.087, 0.119, 0.149) |
| ANN + L2 | (0.208, 0.230, 0.255) | (0.283, 0.293, 0.301) | (0.083, **0.114**, 0.142) |
| ANN + MC dropout | (0.211, 0.231, 0.252) | (0.295, 0.312, 0.328) | (0.089, 0.125, 0.156) |
| Gaussian process | (0.210, 0.231, 0.252) | (0.267, **0.283**, 0.295) | (0.079, 0.117, 0.145) |

Table 16: The $p_{\mathrm{WAIC}_1}$ and $p_{\mathrm{WAIC}_2}$ metric, on our three regression datasets, where the minimum, mean and maximum values are obtained with 10-fold cross-validation.

| | $p_{\mathrm{WAIC}_1}$ (min, mean, max) | | | $p_{\mathrm{WAIC}_2}$ (min, mean, max) | | |
|---|---|---|---|---|---|---|
| Method | Abalone | Wine Quality | Boston Housing | Abalone | Wine Quality | Boston Housing |
| LBBNN | (1.937, 4.486, 14.09) | (6.304, 7.496, 8.617) | (0.037, 0.202, 0.890) | (1.987, 5.206, 20.01) | (6.660, 7.832, 9.642) | (0.038, 0.210, 0.933) |
| LBBNN-MPM | **(0.000, 0.001, 0.002)** | **(0.001, 0.001, 0.003)** | **(0.000, 0.000, 0.000)** | **(0.000, 0.001, 0.002)** | **(0.001, 0.001, 0.003)** | **(0.000, 0.000, 0.000)** |
| LBBNN-LCRT | (0.766, 1.469, 2.228) | (4.978, 5.620, 6.318) | (0.013, 0.128, 0.483) | (0.774, 1.528, 2.595) | (5.112, 5.888, 6.989) | (0.013, 0.133, 0.514) |
| LBBNN-LCRT-MPM | (0.001, 0.001, 0.003) | (0.003, 0.004, 0.006) | **(0.000, 0.000, 0.000)** | (0.001, 0.001, 0.003) | (0.003, 0.004, 0.006) | **(0.000, 0.000, 0.000)** |
| LBBNN-FLOW | (0.563, 1.923, 7.515) | (3.220, 4.202, 6.069) | (0.025, 0.223, 1.025) | (0.568, 2.191, 9.925) | (3.273, 4.407, 7.352) | (0.025, 0.238, 1.157) |
| LBBNN-FLOW-MPM | (0.039, 0.147, 0.430) | (0.171, 0.226, 0.268) | (0.001, 0.012, 0.077) | (0.039, 0.147, 0.432) | (0.172, 0.226, 0.269) | (0.001, 0.012, 0.078) |
| BNN-FLOW | (0.074, 0.237, 0.468) | (0.362, 0.609, 2.210) | (0.003, 0.021, 0.062) | (0.075, 0.238, 0.473) | (0.363, 0.632, 2.415) | (0.003, 0.021, 0.062) |
| BNN | (0.002, 0.007, 0.045) | (0.014, 0.019, 0.039) | **(0.000, 0.000, 0.000)** | (0.002, 0.007, 0.045) | (0.014, 0.019, 0.039) | **(0.000, 0.000, 0.000)** |
| MC dropout | (7.295, 15.42, 44.78) | (18.70, 22.81, 27.87) | (0.344, 1.721, 6.331) | (7.972, 56.82, 430.4) | (20.35, 29.71, 71.86) | (0.397, 2.658, 12.43) |

# E   Resnet-18 architecture for Cifar 10

Table 17: Results on the Cifar 10 dataset with the Resnet-18 architecture, where the min, median, and maximum measurements are obtained from running each method ten times.

| Cifar 10 | Expected calibration error | Negative log likelihood | Density |
|---|---|---|---|
| *Method* | *(min, median, max)* | *(min, median, max)* | *mean* |
| LBBNN | (0.027, **0.037**, 0.067) | (1.350, 1.424, 1.584) | 1.000 |
| LBBNN-MPM | (0.689, 0.763, 0.855) | (50.78, 91.90 245.3) | 0.819 |
| LBBNN-LCRT | (0.053, 0.057, 0.064) | (1.004, 1.039, 1.173) | 1.000 |
| LBBNN-LCRT-MPM | (0.221, 0.243, 0.268) | (1.733, 1.934, 2.117) | 0.386 |
| LBBNN-FLOW | (0.059, 0.062, 0.067) | (0.915, **0.950**, 1.026) | 1.000 |
| LBBNN-FLOW-MPM | (0.347, 0.380, 0.412) | (2.003, 2.230, 2.299) | **0.359** |
| BNN | (0.132, 0.139, 0.146) | (2.799, 3.046, 3.429) | 1.000 |
| BNN-FLOW | (0.091, 0.094, 0.103) | (1.079, 1.097, 1.136) | 1.000 |
| ANN | (0.120, 0.207, 0.213) | (3.444, 4.173, 5.029) | 1.000 |
| ANN + BN | (0.195, 0.200, 0.208) | (1.822, 1.930, 2.133) | 1.000 |

