# OpenReview forum: "Sparsifying Bayesian neural networks with latent binary variables and normalizing flows"
_TMLR — Accepted by TMLR_

### Review · Reviewer_NqKg · 2024-06-01

**Summary Of Contributions:**

I served as a reviewer (K8JC) for this paper's first submission; see my first review at https://openreview.net/forum?id=t3OGrWRUve&noteId=0vpySh1cZ6. As the main focus and arguments of the paper haven't changed, I repeat my earlier summary below.

The authors propose modelling and optimisation improvements to latent binary Bayesian neural networks (LBBNN): Bayesian neural networks with a spike-and-slab prior and a variational spike-and-slab posterior on the weights. The authors propose using a conditionally factorised posterior approximation instead of a fully factorised one to improve the expressivity of the variational posterior and use a normalising flow to model the posterior distribution of the newly introduced conditioning variable. They perform some toy experiments on MNIST-type datasets and synthetic data.

**Audience:**

Yes

**Claims And Evidence:**

No

**Requested Changes:**

Necessary:
 - As I outlined in the weaknesses section, please clarify the claimed contributions and point to precise advances/results to back them up.
 - Please clarify the claim about the convergence I note in the weaknesses section.
 - The bibliography style needs to be more consistent; for example, authors' names appear inconsistently; please fix it.
 - Figures 6-8: Some data points fall outside axes, so please clip these or enlarge the axes.
 - Figure 9: x-axis labels are missing; please fix this.

Nice to have:
 - Any improvement in the writing that would address the issues I outline regarding it in the weaknesses section above.

**Strengths And Weaknesses:**

## Strengths
The paper's strengths have stayed the same. The authors' model and inference procedure are a natural extension of previous methods. Their method description in Sections 2-4 is clear and easy to follow, and they have also further polished it since the last submission. Moreover, the authors have conducted a significant amount of ablation studies to verify the effectiveness of their proposed changes.

## Weaknesses
Having carefully re-read the paper, I think the content of the paper is mostly acceptable. However, I have some issues with the claimed contributions. Concretely, the authors claim four things:
1. "extending the class of variational distributions to normalising flows allowing modeling dependencies;" - This claim is too vague. It is unclear what class they extend, what dependencies they are modelling, and what the extension's purpose is.
2. "improvements of computational efficiency in LBBNNs through the use of the local reparametrisation trick (Kingma et al., 2015);" - This claim is again too vague; please specify concrete metrics and results.
3. "demonstrating improvements in predictive power, sparsity, and variable selection through experiments on real and simulated data;" - The relevance of these contributions is unclear.
4. "demonstrating robust performance in uncertainty quantification through the expected calibration error for classification and Pinball loss for regression." - All pinball loss regression-related content is in the appendix. If the authors want to claim this, then this content should appear in the main text.

Besides this, I am concerned about one technical claim in Section 5.2. The authors claim, "When we increase the amount of data, we can see that the Bayesian networks gradually get more certain about predictions, and the entropies (as desired) start to converge towards the data-generative one." At best, this statement is meaningless and at worst, it is misleading. If the authors wish to make such a claim, they must provide a formal statement and proof of the result.

If the authors address my concerns above, I believe it should conform to TMLR's acceptance criteria and could be published.

Having said that, I am sad to say that despite the significant experimental effort on the authors' part, the paper's presentation is severely lacking.

First, the purpose of Section 1.1 is unclear to me. In my review of the previous submission, I complained that the introduction read like a literature review, so the authors wrote a better introduction. However, instead of removing their previous introduction, they moved it to Section 1.1 and renamed it "Literature Background." I have two issues: 1) The introduction (which is reasonably well-written) and Section 1.1 are very similar in content. Furthermore, Section 1.1 has been repurposed as the literature background in a cursory way, as there are entire sentences (!) in it that match verbatim with sentences in the introduction, including the first sentence! 2) Section 1.1 still contains a lot of unnecessary content that does not help contextualise the authors' work. For example, it reviews early neural networks and gradient descent; these sections should be removed.

I have similar issues with the description of the experiments in Section 5, which is a whopping nine and a half pages long. While most of the content is technically fine (apart from the one claim I take issue with above), the presentation leaves much to be desired. It contains long arguments, with the experiment setup, method description, and interpretation of results blended together, making it difficult to follow. I believe that with some effort to streamline the arguments, the authors could cut the length of Section 5 in half at least.

---

> ### Author Response · Authors · 2024-09-13
> **Response**
>
> We thank the reviewer for the constructive feedback and dedication. Below, we shall answer on various aspects   and clarify our points. The revised text in purple in the resubmitted manuscript.
>
> **The purpose of Section 1.1:**
> We fully agree on that this section should have been written in more accordance with the new introduction. We have now removed a large part of this section which overlapped with the introduction and incorporated the remaining part into the introduction itself.
>
> **The bibliography style needs to be more consistent; for example, authors' names appear inconsistently; please fix it.**
>
> Authors’ names and some capitalization errors have now been rectified. Thanks for pointing this out.
>
> **Figure 9: x-axis labels are missing; please fix this**
>
> This experiment has been removed.
>
> **Figures 6-8: Some data points fall outside axes, so please clip these or enlarge the axes.
> Done, axes have been enlarged.
> Please clarify the claim about the convergence I note in the weaknesses section.**
>
> We changed the wording on this, also removed the entire section with mixtures of Gaussian experiments as that simulation was both too long and rather brought confusion to the other reviewer than additional insights. However we added additional variable selection simulations, where all images are saved in high resolution svg to provide the best quality.

---

> ### Author Response · Authors · 2024-09-13
> **Continued**
>
> **As I outlined in the weaknesses section, please clarify the claimed contributions and point to precise advances/results to back them up.**
>
> We have removed the claim contribution regarding pinball loss, as we think these results are not important enough to appear in the main text, and we do not want to make it any longer. Additionally, we have slightly rewritten the other claims as per the reviewers request. Further, we added ECEs and NLLs for all image datasets.
>
> **I believe that with some effort to streamline the arguments, the authors could cut the length of Section 5 in half at least.**
>
> This section is now much shorter and more readable, after removing the entropy experiment. We also added NLLs and ECEs for the image classification experiments in the main part of the paper. But if you recommend to move some of them to the Appendix, we shall be happy to do that.
>
>
> We hope the reviewer finds our responses sufficient and we shall be happy to continue the discussion on openreview if there are further comments or requests.

---

> > ### Comment · Reviewer_NqKg · 2024-09-16
> >
> > I thank the authors for their responses. I think they have done a good job at improving the introduction and the experiments section. I am now ready to recommend acceptance of the paper, with a couple of minor revisions:
> >  - The text in Figures 4, 5 and 6 are quite small and difficult to read at 100% zoom; please increase the font size to be roughly equal to the font size in the main text (perhaps *slightly* smaller)
> >  - Please split each of the two panels in Figure 6 into three separate ones separated by whitespace with individual axes, as currently the plots look misleading. They look like a continuous signal, whereas in reality it is the same experiment with different set-ups concatenated next to each other three times. Especially the x-axes on these panels are bad data presentation practice.

---

> > > ### Author Response · Authors · 2024-09-17
> > > **response from the authors**
> > >
> > > We thank the reviewer for the commitment at constructive feedback throughout the whole review process. And we thank the reviewer for the reccomendation. We assure that upon acceptance the requested minor presentational changes in the graphs will be made for the camera ready version of the paper.

---

> > > > ### Author Response · Authors · 2024-10-25
> > > > **updates in the camera ready version**
> > > >
> > > > We have updated the Figures in the camera ready version. Figures 4 and 5 are made larger with larger font sizes. Figure 6 in the main part of the paper and Figures 7 - 9 in the Appendix are rearranged to have individual axes and also larger fonts. Thank you once again for the thorough review in both of the rounds.

---

### Review · Reviewer_F54M · 2024-06-10

**Summary Of Contributions:**

In this paper, they consider two extensions of Latent Binary Bayesian Neural Networks (LBBNN): the first is the use of the local reparametrization trick, and the second is the use of normalizing flows. They provide experimental results for their methods.

**Audience:**

No

**Claims And Evidence:**

Yes

**Requested Changes:**

- It would be beneficial to explain other sparse BNN papers (at least [1], [2], [3], [4]) and discuss how the proposed method differs from them.

- It would be beneficial to experimentally compare with 2-3 of other sparse BNN methods (e.g., [1], [2], [3], [4]).

- It would be beneficial to include experiments using more challenging datasets (e.g., CIFAR10, CIFAR100, SVHN) and more complex models (e.g., ResNet18). Additionally, it is recommended to report the Negative Log-Likelihood (NLL) or Expected Calibration Error (ECE) results for image classification.

**Strengths And Weaknesses:**

# Strengths

- To make their ideas easy for readers to understand, they effectively explained their ideas through many figures and explanations.

- They present various types of meaningful experimental results.

# Weaknesses

- I believe the proposed method lacks novelty since it combines existing methods. However, if the experimental evidence strongly supports the proposed method, this may not be a significant issue.

- There is insufficient discussion on other sparse BNNs. Specifically, there are papers published in top conferences such as [1], [2], [3], and [4]. These should be mentioned, and the proposed method should explain how it differs from them.

- I am concerned that the baseline is insufficient. Currently, the only sparse baseline method used for comparison is LBBNN. Given the lack of novelty in this paper, it is essential to compare its performance with other methodologies.

- I am also concerned that classification experiments were conducted only on the MNIST dataset (and its variations) and the LeNet model, which are too simple. Also, there is no report on uncertainty measures in the manuscript.

[1] Molchanov, D., Ashukha, A., & Vetrov, D. (2017, July). Variational dropout sparsifies deep neural networks. In International conference on machine learning (pp. 2498-2507). PMLR.

[2] Deng, W., Zhang, X., Liang, F., & Lin, G. (2019). An adaptive empirical Bayesian method for sparse deep learning. Advances in neural information processing systems, 32.

[3] Bai, J., Song, Q., & Cheng, G. (2020). Efficient variational inference for sparse deep learning with theoretical guarantee. Advances in Neural Information Processing Systems, 33, 466-476.

[4] Li, J., Miao, Z., Qiu, Q., & Zhang, R. (2024). Training Bayesian Neural Networks with Sparse Subspace Variational Inference. arXiv preprint arXiv:2402.11025.

---

> ### Author Response · Authors · 2024-09-13
> **regarding lack of novelty**
>
> We thank the reviewer for the constructive feedback. Below, we shall answer on various aspects   and clarify our points. The revised text is in purple in the resubmitted manuscript.
>
> **I believe the proposed method lacks novelty since it combines existing methods. However, if the experimental evidence strongly supports the proposed method, this may not be a significant issue.**
>
> While we agree with the approach being a combination of two existing ideas, it is very common in variational inference to propose more flexible variational approximations and investigate them. In this case not modeling dependencies in the original LBBNN was seen as a disadvantage, which we wanted to resolve in this work.
> We further believe the paper motivates well why we need BNNs that incorporate both structural and parameter uncertainty (although more details can be naturally found in the original LBBNN papers that introduced the model), while allowing for more flexible variational distributions. Our experimental results do sanity checks and demonstrate that the proposed method is robust, on par or better than the most closely related methods with mean field approximations, in terms of predictive power, better in uncertainty handling and somewhat better when it comes to variable selection with correlated covariates. The claims are based on a clear treatment effect and are valid at least for the sets of data we addressed. Also all three reviewers from the first round found claims and evidence supported: https://openreview.net/forum?id=t3OGrWRUve&referrer=%5BAuthor%20Console%5D(%2Fgroup%3Fid%3D TUMBLR%2FAuthors%23 your-submissions.
> We believe that we have also motivated why we are not trying to obtain state of the art results, and also why we do not include or use common SOTA hacks found in the BNN literature such as posterior tempering or hypertuning of the parameters in order to improve predictive performance. Much of the modern ML research is chasing state-of-the-art based on massive tuning, heuristic hacking of predictive performance, biased experimental design without clear treatment effects of what actually contributes to getting these state-of-the-art results (the method or the tuning or both). We believe this to be somewhat poor empirical science practice and backup this claims with a recent but important position paper [1a], where all pitfalls of a common practice of  “beating baselines and achieving SOTA on a bunch of data” is criticized. In that sense, we rather follow guidances of TMLR https://jmlr.org/tmlr/acceptance-criteria.html, which are also highlighted in [1a], and simply claim modest improvement of inference for LBBNN under correlated data settings and empirically demonstrate it with clear “treatment” effects for the suggested improvements in a predefined experimental design that did not involve any hyper-tuning of the performance. We also only include comparisons that would allow us to evaluate direct “treatment” effects within our design of experiments as demonstrated in Figure 3. We try to not be biased in favor of our method by means of doing this. Although SOTA hacking may be possible for our method, we do not want to engage ourselves in this practice, hence we do not want to make any SOTA related claims. We neither claim stellar novelty, but rather an improvement of approximations for LBBNN. We believe our experiments demonstrate the improvements we claim in the paper for when the new flow based approximations are expected to outperform the mean-field results that should hopefully meet the TMLR criteria.
> We leave experimentation and comparisons with “competing methods” beyond the clear design from Figure 3 (i.e. when the direct treatments are not available) to neutral comparison studies, where the authors of “new” methods (us in this case) would not take advantage of tuning their method best (either on purpose or simply due to knowing their methods best) and “beating all strong baselines“. See [2a].
>
> [1a] Herrmann, Moritz, et al. "Position: Why We Must Rethink Empirical Research in Machine Learning." Forty-first International Conference on Machine Learning.
>
> [2a] Boulesteix, Anne-Laure, Sabine Lauer, and Manuel JA Eugster. "A plea for neutral comparison studies in computational sciences." PloS one 8.4 (2013): e61562.
>
> [3a] Jiri Hron, Alexander G de G Matthews, and Zoubin Ghahramani. Variational Gaussian dropout is not Bayesian. arXiv preprint arXiv:1711.02989, 2017.

---

> > ### Author Response · Authors · 2024-09-13
> > **Discussion of other sparse BNNs**
> >
> > **There is insufficient discussion on other sparse BNNs. Specifically, there are papers published in top conferences such as [1], [2], [3], and [4]. These should be mentioned, and the proposed method should explain how it differs from them.**
> >
> > This is a good point, we have now included (in the introduction) some discussion around these methods and how they differ. Also a lot of discussion about motivation of LBBNN and other sparse methods can be found in the original paper introducing LBBNN as answered to the reviewer above.

---

> ### Author Response · Authors · 2024-09-13
> **more challenging datasets**
>
> **It would be beneficial to include experiments using more challenging datasets (e.g., CIFAR10, CIFAR100, SVHN) and more complex models (e.g., ResNet18). Additionally, it is recommended to report the Negative Log-Likelihood (NLL) or Expected Calibration Error (ECE) results for image classification.**
>
> We added NLL and ECEs for all studied image datasets and methods. In practically all cases, one of the LBBNNs achieved the best performance in terms of uncertainty handling for image data. In most of these cases, it was associated with the flow-based Variational Inference.
> Also, we, as asked by the reviewer, did computation of ECEs and NLLs for CIFAR 10 and a version of Resnet 18 without batch norm for the set of baselines we address (plus we added an ANN with batchnorm). The results on ECEs and test negative log likelihood are added to the Appendix. The proposed LBBNN-FLOW and LBNN-LCRT seem to work there quite well in the sense of NLLs, sparsity and ECEs.

---

> ### Author Response · Authors · 2024-09-13
> **Regarding insufficient baseline**
>
> **I am concerned that the baseline is insufficient. Currently, the only sparse baseline method used for comparison is LBBNN. Given the lack of novelty in this paper, it is essential to compare its performance with other methodologies.**
>
> Firstly, the lack of novelty is a subjective thing and TMLR https://jmlr.org/tmlr/acceptance-criteria.html seems to not be too critical to it.
> Secondly, we would like to note that some remote but related sparsity inducing methods were already extensively compared to the original LBBNN already in the early version of it https://arxiv.org/abs/1903.07594, including the concrete dropout, horseshoe priors based models and mixtures of Gaussians priors, corresponding to continuous spike and slab. Where the baseline LBNNN performed quite well, although that paper does not have direct treatment effects and it is not a neutral comparison study [2a], which would be required for more unbiased conclusions.
>
> In this paper we are not introducing the LBBNN, but we are interested in improvements wrt LBBNN’s standard mean-field inference. In a carefully designed experimental graph from Figure 3, direct “treatment” effects are incorporated allowing for reasonable comparison of which of these “treatments'' improve sparsity, predictions, and/or uncertainty handling (the latter for tabular data). We hence did not want to add clearly different methods that can not give direct “treatment” edges in Figure 3. We follow TMLR guidance to which we already referred, in what we claim and what evidence we use to support our claims.
> Yet, further we discuss why comparing to methods suggested by the reviewer does not make sense in our experiments in the light of the important position from [1a]. We would still believe a neutral comparison study [2a] of all of the aforementioned papers suggested by the reviewer could be an important empirical contribution, but since we are introducing a new method here, neutral comparison is impossible as we are biased to knowing our method better than other methods. The need for a neutral comparison study is now clearly emphasized in the discussion section.
>
> As we mention in our experimental design, we want to avoid using ad-hoc tricks to push to state of the art at all costs, and we want to explicitly avoid comparing to methods that use very different approaches and in addition use many hacks and tricks for training differently in each data set, both not allowing us to have reasonable “treatment” effects in our experimental design to avoid making comparisons between fundamentally different entities. Also, since we only claim how inference on LBBNN can be improved by more flexible variational approximations, those state-of-the-art hacking experiments would not be useful to confirm our claim. The current experimental design in Figure 3 of our paper is chosen to have direct “treatment” on every edge.
> Further, we avoid any overturning in our experiments: we do random initialization only, do not use tempering to reduce the so called “cold posterior effect”, and we only choose the median probability model for sparsification, although other thresholds on marginal inclusion probabilities could be tuned to achieve for a specific data somewhat better sparsity or accuracy.  Same applies to learning rates of the used first-order stochastic optimisation tools. We leave potential SOTA hacking outside the scope of our work due to it being problematic as an empirical research tool [1a].

---

> > ### Author Response · Authors · 2024-09-13
> > **continued**
> >
> > Regarding [1], we do mention variational dropout briefly, we now added further discussion on the impressive sparsity levels with impressive accuracies it can produce. But we do not think it makes a lot of sense to include it in our experimental design, as using the log-uniform prior combined with common likelihood functions induces an improper posterior (see [3a]), therefore we believe the results are not only too remote from our approach, but also should not be considered valid from a Bayesian inference perspective. More importantly, the paper [1] while producing impressive results does a lot of experiment specific tricks and tuning that would be problematic in the light of [1a], again, not allowing us to see the “treatment” effect of whether the tricks or the methodology contributes to the reported results. E.g. in section 5.4 of [1], VGG like network is considered with pre-activation batch normalizations, which are not quite justified to be combined with the variational Bayes in [1], and also combined with binary dropout without justification, furthermore pretrained architectures were used by the authors in [1] is some experiments (but not other), yielding impressive results, but not allowing us to consider the method in a neutral confirmatory and conclusive experimental design. For other tricks in [1], see their section 5.1 general empirical observations, where many ad-hoc tricks for sparsification and optimization are discussed, including tempering the posterior, choosing thresholds for pruning and tricks for initialization. As we use different architectures, we are actually not sure which tricks should be used to make [1] work well in a pre-defined experiment for our architectures.
> >
> > Likewise for [2], while this method shows without doubts impressive results on compression, we do feel that it is quite dissimilar to our method thus not allowing to get the “treatment” effect: It uses a different prior distribution based on a continuous spike and slab with L1 regularization in one component through Laplace and L2 in the other component through a Gaussian, inference is not done with variational Bayes, but rather with a mix of Empirical Bayes (to tune priors), hybrid Bayes (using point estimates for some parameters), and tempering of MCMC, and  finally sparsity is obtained by magnitude based pruning with tunable thresholds, thus not allowing to place these changes in a sensible way into Figure 3 in our design of experiments. Including [2] will neither contribute to our claims in the submitted that we want to empirically verify.  Yet, we add the discussion on [2] in the revised paper.
> >
> > We were not aware of [4] at the time of writing, and this is very interesting work and impressive results too. However there are also some big differences here to our work, most notably that the inclusion parameters are not part of the Bayesian framework, but rather are pruned (and reintroduced back) similar to frequentist methods. We feel that this deviates quite strongly from our approach, and therefore comparing results is futile and would not fit the design in Figure 3. Moreover, just like in [1] and [2] various tricks are used in different experiments in [4], including KL warmup, strategies for initialization, etc, making it a bit difficult for our pre-specified design with direct “treatment” effects. Additionally, this method is using a tempered posterior distribution. While this is common in the Bayesian DL literature to reduce the so called “cold posterior effect”, it does not have a solid justification for why it provides better empirical results than the true Bayes posterior used as a target.
> >
> > Finally, we also do not think it is worthwhile to compare additionally to [3] for a different reason than other suggested methods.  The method in [3] is essentially the same (except for the choice of specific prior inclusion probabilities) as the baseline LBBNN that first appeared earlier than [3] in https://arxiv.org/abs/1903.07594. And while [3] refers to https://arxiv.org/abs/1903.07594 (which proves the authors were aware of it), yet [3] does not compare to https://arxiv.org/abs/1903.07594 in their experiments, where the comparison could have been indeed reasonable given the marginal changes in [3] compared to the original LBBNN, while [3] still compares to other quite unrelated methods without direct “treatment” effects possible. We do not think the issues of design in [3] is something that we are supposed to fix in this paper. However, we did refer to [3] already in the reviewed paper, and we still gladly discuss it in our revised  paper.

---

> > > ### Author Response · Authors · 2024-09-13
> > > **continued**
> > >
> > > To summarize, we feel that LBBNN (and the closely related Carbonetto/Stephens methods) as well as some methods in their direct neighborhood (with clear “treatments” on edges) as shown in Figure 3 are most suitable for comparing results on sparsity, as we are trying to demonstrate that introducing correlations in the variational posterior distribution improves over mean-field. We again emphasize that the goal is not to obtain state of the art predictive performance results on sparse (Bayesian) neural networks, we do not claim it and in fact want to avoid such claims in general following [1a], and therefore we do not think it is reasonable to compare against methods that differ quite strongly from ours and that use massive tuning when reporting their results. However we think that the above points should be included and emphasized more strongly in our manuscript. We do agree that there should also be more discussions around the other sparse Bayesian methodologies, which we included in the paper. Finally, we believe that a proper neutral comparison study is of interest in the future.
> > >
> > > We hope the reviewer finds our responses sufficient and we shall be happy to continue the  discussion on openreview if there are further comments or requests.

---

### Review · Reviewer_9rKf · 2024-08-29

**Summary Of Contributions:**

This paper considers a class of deep probabilistic models known as latent binary Bayesian neural networks (LBBNN) wherein a binary variable $\gamma_{ij}^{(l)} \in$ {0,1} is associated with each weight $w_{ij}^{(l)} \in \mathbb{R}$ in the neural network, such that $w_{ij}^{(l)} =0$ if $\gamma_{ij}^{(l)} =0$. The exact posterior $P(\boldsymbol{W}, \boldsymbol{\Gamma} \mid -)$ in these models is highly intractable due in part to the combinatorics of the binary variables.

The paper adopts a variational approach to inference. Previous work by Carbonetto & Stephens (2012) showed that mean-field variational approximations to the posterior of spike-and-slab Bayesian logistic regression tend to perform poorly at variable selection when the covariates are correlated. The paper thus embraces the hierarchical variational inference approach of Ranganath et al. (2016), wherein the variational distribution is augmented with a latent variable $\boldsymbol{z}$, becoming $q(\boldsymbol{W}, \boldsymbol{\Gamma} \mid \boldsymbol{z}) q(\boldsymbol{z})$, which induces dependence among the weights. The paper further combines this with normalizing flows (Rezende & Mohamed, 2015), modeling $q(\boldsymbol{z})$ flexibly as coming from a composition of invertible transformations.

To tractably fit the proposed model with the proposed hierarchical + normalizing flows variational family, the paper further introduces an approximation based on the local parameterization trick (LCRT) of Kinga et al. (2015). The final fully specified proposed approach is referred to as LBBNN-FLOW. The paper then also proposes a simplified version of the approach that assumes the $\boldsymbol{z}$ are all equal to 1, which it refers to as LBBN-LCRT. (Both approaches involve the LCRT approximation.)

The experiments show that:
- The proposed approach performs better than the original logistic regression baseline by Carbonetto & Stephens (2012) on a variable selection task
- The proposed approach has higher predictive uncertainty than a non-Bayesian NN
- The proposed approach obtains comparable classification accuracy on *NIST data sets to non-sparse models

**Audience:**

Yes

**Claims And Evidence:**

No

**Requested Changes:**

The paper needs to be significantly revised to say clearly why LBBNNs are motivated, and in what aspects the proposed model improves over current practice. I am very open to any of the above-mentioned (or other) aspects, but the paper needs to commit to one (or more) and then provide experiments that clearly show improvement along those dimensions.

I mentioned some of these possible experiments in the above section but here they are again:
- If the paper wants to claims that LBBNNs improve in terms of variable selection, there needs to be a much more rigorous set of experiments that substantiate this claim, which include many simulated data sets, at different levels of covariate correlation.
- If the paper wants to claim that LBBNNs improve in terms of predictive performance, than we need to clearly see evidence of that.
- If the paper wants to claim that LBBNNs improve in terms of uncertainty quantification, then the paper needs to defend the implicit claim that higher predictive entropy is actually good, and correct in the given setting.  Moreover, any claims of calibration must be defended, theoretically or empirically.
- If the paper wants to claim an improvement in efficiency, we should see performance as a function of wall-clock time.

**Strengths And Weaknesses:**

## Strengths

There is general interest in the Bayesian community in sparse spike-and-slab models and how to perform tractable posterior inference in them. This paper combines several difference modern frameworks / approaches (i.e., hierarchical VI, normalizing flows, BNNs) to provide an approach for fitting deep sparse models. Combining these frameworks involves some tailored derivations, which are correct so far as I can tell, and which may be of broader interest.

## Weaknesses

The paper does a poor job of motivating the proposed model, and fails to empirically substantiate its claimed advantages. The paper is vague throughout about why it is interested in developing LBBNNs. At various points, the paper suggests that LBBNNs might be motivated by the following aspects:
- improved variable selection (particularly when covariates are correlated)
- improved predictive performance
- improved uncertainty quantification
- improved computational efficiency
- improved memory efficiency

I think any one of these aspects would be interesting and motivating. However, the paper does not commit to any one of them being the main motivation, and the experiments do not convincingly show a major advantage along any one of them. I'll walk through each aspect below.

**Variable selection**: The paper includes one experiment on this task which involves one simulated data set. The proposed approaches are compared to the logistic regression baseline of Carbonetto & Stephens (2012). Both proposed approaches have better TPR and FPR than the baseline on the one simulated data set. This is a good result, but it may be due entirely to properties of the single synthetic data set. The paper should simulate many synthetic data sets, and report error bars across them. Moreover, the claim of the paper is that the proposed approach will work better than the baseline as the covariates become more correlated. This claim would be substantiated by a plot showing that performance relative to the baseline improves as the covariates become more correlated in a series of simulated data sets.

**Uncertainty quantification**: The paper shows that a baseline method based on a BNN with dropout "has high [predictive] certainty most of the time" while the baseline does not. This is measured using the entropy of the predictive distribution across a grid of covariate values. However, the paper does substantiate the implicit claim that higher certainty is bad in this context. Indeed, looking at the plots in Figures 6-8, I actually think Dropout looks the _best_ given how separated the training data are. It is trivial to create a new method that has higher entropy in its predictive distribution; but is that necessarily good? The paper isn't clear about that.

In the discussion, the paper also claims that the "calibration of uncertainties [...] is similar with a slight advantage of the proposed approach". However, I did not see any theoretical statement about the calibration of the proposed approach's predictive probabilities, nor any experiment which tests it.

**Predictive performance**: The experiments (at least as I understand them) do not show a major improvement in predictive performance. The paper says that the performance of the sparse models are "comparable" to the dense ones. This is interesting, but only so far as there is another reason to prefer the sparse models.

**Computational / memory efficiency**: The paper is not clear whether the proposed approach improves in terms of efficiency over a BNN baseline. If it does in fact improve, a plot showing performance as a function of wall-clock time should be included.

---

> ### Author Response · Authors · 2024-09-13
> **Responses of the authors**
>
> We thank the reviewer for constructive feedback. In the revised paper, we made an effort on being more clear and consistent with what we claim and how we support the claims, see the bullet points in the end of sec 1 as well as with new results and within the discussion section. The revised text in purple in the resubmitted manuscript.
>
> **The paper does a poor job of motivating the proposed model, and fails to empirically substantiate its claimed advantages. The paper is vague throughout about why it is interested in developing LBBNNs.**
>
> We would like to humbly draw the reviewer’s attention to the fact that this paper does not introduce LBBNN, which appeared in [4a, 5a], and hence discusses less of the advantages of the model (although some of the advantages with LBBNN are discussed in the introduction, page 2, middle paragraph). Yet, the paper introduces new inference techniques to approximate the posterior distribution of inclusion indicators and the weights. Thus the main point we show here is that the new inference techniques work as compared to the original mean-field approximation without LCRT or normalizing flows in both identifying model uncertainty (variable selection experiments), and predictive sparsity, uncertainty handling, and accuracy.
>
> **I think any one of these aspects would be interesting and motivating. However, the paper does not commit to any one of them being the main motivation, and the experiments do not convincingly show a major advantage along any one of them. I'll walk through each aspect below**
>
> We intentionally do not want to concentrate on one aspect but rather study the behavior of the method in terms of all of the points mentioned above (model uncertainty, sparsity, predictive uncertainty and accuracy) and try to fairly show not only the benefits, but also cases where the improvements are marginal or even absent, following criticism of empirical research in ml and some of the suggestion to improve it from  [1a]. Our main purpose is to demonstrate the potential benefits in improving the original algorithm for fitting the LBBNN model by the local reparameterization trick and the use of normalizing flows. For all the methods compared, further improvements can possibly be achieved by fine tuning different hyperparameters, but this has not been performed in order to obtain a more fair comparison between the methods. We are therefore not interested in state of the art as we clearly write in our design of experiments (although we are close to them on CNN architectures for -NIST experiments). We are rather interested in seeing the behavior of the method in various scenarios and doing sanity checks (also see responses to other reviewers). All three reviewers from the first round found our claims supported by the data: https://openreview.net/forum?id=t3OGrWRUve&referrer=%5BAuthor%20Console%5D(%2Fgroup%3Fid%3DTMLR%2FAuthors%23your-submissions. Yet, of course reasonable improvements are always possible and useful. Below, we shall go step by step to clarify the aspects listed by the reviewer. As requested by the other reviewer, calibrations are added to all image experiments. We also conducted 4 additional simulation studies with various correlation structures, correlation levels, and noise levels, as you requested. The answers to all of your concerns are given is separate messages below.
>
> References:
>
> [1a] Herrmann, Moritz, et al. "Position: Why We Must Rethink Empirical Research in Machine Learning." Forty-first International Conference on Machine Learning.
>
> [2a] Sommerfelt, Philip Sebastian Hauglie, and Aliaksandr Hubin. "Evolutionary variational inference for Bayesian generalized nonlinear models." Neural Computing and Applications (2024): 1-18.
>
> [3a] Høyheim, Eirik. Improving sparsity and interpretability of latent binary Bayesian neural networks by introducing input-skip connections. MS thesis. Norwegian University of Life Sciences, 2024.
>
> [4a] Hubin, Aliaksandr, and Geir Storvik. "Combining model and parameter uncertainty in Bayesian neural networks." arXiv preprint arXiv:1903.07594 (2019).
>
> [5a] Hubin, Aliaksandr, and Geir Storvik. "Sparse Bayesian neural networks: Bridging model and parameter uncertainty through scalable variational inference." Mathematics 12.6 (2024): 788.

---

> > ### Author Response · Authors · 2024-09-13
> > **improved variable selection (particularly when covariates are correlated)**
> >
> > **improved variable selection (particularly when covariates are correlated**
> >
> > We conducted one study using a complicated correlation structure between the binary and continuous covariates already in the reviewed paper. Yet, we agree that this was somewhat limited. To be less trivial, we added 4 new simulation studies with 10 levels of non-trivial correlations strengths between the true and false positive covariates and repeated for 3 noise levels each, where all methods are run 20 times per settings, thus resulting in 2400 new runs per method involved. No additional tuning was done and exactly the same settings for all methods were used as in the submitted paper. The new simulation study based on data from Section 5.1 is added to Section 5.1.1, where the results are consistent with the original study, while other three studies are added to the Appendix to avoid further increasing the main paper, which would be in conflict with the requests of one of the other reviewers. There, for the QTL correlation structure flows are the best, while for the two other structures (simrel and our own), all methods perform similarly and without uniform “winners”, with CS doing, however, better in the simrel scenario.
> >
> > Moreover, recently, [2a] (simulation study 1 and 2) and [3a] (section 4.1, 3 simulation studies) used our method to compare the quality of variable selection using both flows and mean-field approaches, where correlation level for one false positive was changed to exactly one of the data generative covariates. Those studies systematically showed better performance of the flows in these simple correlated settings compared to LCRT. This discussion is added to the revised paper too.

---

> ### Author Response · Authors · 2024-09-13
> **improved predictive performance**
>
> **improved predictive performance**
>
> First of all, we would like to clarify that we never claimed overall improved predictive performance with respect to traditional methods from our experimental design (ANN, ANN-L2, BNN, and BNN-FLOW), but rather similar performance with an advantage of sparsity. What we did claim, however, is that within inferences for LBBNN, the approximation of the posterior method with flows typically and on the studied sets of data results in slightly improved performance and sparsity. We tried to make it more clear in the revision. In all of the image classification experiments flow based inference allows to obtain the best median accuracy among the LBBNN models for all experiments under both dense and sparse predictions. Moreover, in 6 out of 12 cases the lower bounds of accuracies of LBBNN with flow inference is higher than the upper bounds of LBBNN with mean-field based inferences. Further, for the UCI datasets reported in the Appendix the flow inference results in the best median predictive performance in circa half of the classification problems, but for regression LCRT based mean field inference performed slightly better. We have now changed the claim accordingly. We do not claim statements about statistical significance as only 10 runs per settings per method were used. Yet in many cases the lower bounds of the accuracy of the proposed method with flows is above the upper bounds achieved by the mean-field and LCRT.

---

> ### Author Response · Authors · 2024-09-13
> **improved uncertainty quantification**
>
> **improved uncertainty quantification**
>
> We would like to humbly draw the reviewer’s attention that already in the reviewed paper, we had ECEs and Pinball losses reported for the UCI datasets (that could have been missed as they were only reported in the Appendix), where one of the LBBNN fitting methods archives the best median performance in 6 out of 9 addressed data sets. Now, in the revised manuscript, for the image datasets, we demonstrate that one of LBBNNs is achieving the best calibration performance in practically all of the cases. Of course the conclusions are limited to our experimental scenarios.
>
> **Uncertainty quantification: The paper shows that a baseline method based on a BNN with dropout "has high [predictive] certainty most of the time" while the baseline does not. This is measured using the entropy of the predictive distribution across a grid of covariate values. However, the paper does substantiate the implicit claim that higher certainty is bad in this context. Indeed, looking at the plots in Figures 6-8, I actually think Dropout looks the best given how separated the training data are. It is trivial to create a new method that has higher entropy in its predictive distribution; but is that necessarily good? The paper isn't clear about that.**
>
> The goal of this experiment was to show the uncertainties graphically but also show that following  Bernstein–von Mises principles the uncertainty reduces for LBBNN/BNN as the signal/dataset size increases, while this is not the case for dropout. Indeed this corresponds to potentially bigger (than dropout) uncertainty for smaller datasets/signals, while smaller than (dropout) uncertainties for bigger datasets/signals. But since the experiment is long and also criticized by the other reviewer for that matter, we decided to just remove it. Instead, we added negative log-likelihoods (NLLs) and Expected calibration errors (ECEs) for all image datasets in addition to the ECEs and pinball losses on UCI datasets reported already in the reviewed manuscript.

---

> ### Author Response · Authors · 2024-09-13
> **improved computational efficiency and improved memory efficiency**
>
> **improved computational efficiency**
>
> We here need to differentiate between training and testing complexity.
> The training complexity with the first order optimization is linear in the number of data points in a minibatch size, and linear in number parameters, and finally linear in the number of samples from parameters for the obtained the estimates of the gradient. LRCT improves computational efficiency of training LBBNN as we are sampling directly from the neurons, but then FLOWS introduce additional trainable parameters, which in turn will slow the training down and can become even slower than standard LBBNN if flows have too many more parameters. We would like to humbly point the reviewer’s attention to that already in the reviewed manuscript we actually have casually (due to running everything on a shared node) reported the average compute times per epoch for LBBNNs on -NIST data (that was in the end of results section). Yet,  as we share the computational nodes with the whole department and cannot in general  guarantee equal loads for all methods.  Now, we added theoretical training costs. Training efficiency theoretical results are summarized  in Table 8 in the Appendix.
>
> For testing/prediction, sadly no speed up in practice can  be obtained as of now because our GPUs and Torch tensors to the best of our knowledge cannot handle sparse matrices more efficiently than dense matrices, but possible memory efficiency are discussed below.
>
> **improved memory efficiency**
>
> At the training stage we do not get memory efficiency, but rather have to store an additional parameter per weight plus the flow parameters. The same concerns trained non sparsified models. But for storing the sparsified with MPMs models, one saves a lot assuming that 64 bits is used to store the active weights, but only 1 bit is used for whether the weight is pruned. Memory efficiency theoretical results (under some assumptions on the types used for storing them) are summarized  in Table 8 in the Appendix.
>
> We hope the reviewer finds our responses sufficient and we shall be happy to continue the discussion on openreview if there are further comments or requests.

---

### Author Response · Authors · 2024-09-13
**The revision is now uploaded**

We would like to thank the editor and the reviewers for constructive and helpful feedback.
We have now finalised the revision and uploaded the revised manuscript.
Changes from the original submission are in purple colour. Below, we provide specific responses to the reviwers' comments.
We will be very happy to continue the discussion on open review.

---

### Author Response · Authors · 2024-10-25
**The camera ready version is now uploaded**

We thank the action editor for his decision. We would also like to thank the reviewers in both rounds for their valuable feedback and comments that doubtlessly improved our paper. Minor revisions are done in the uploaded version and the Figures are accordingly updated.

---

> ### Comment · Action_Editor_2c4u · 2024-10-28
> **Accept**
>
> The authors fixed the figures as requested by the reviewer. The paper is now ready for publication.

---

### Decision · Action_Editor_2c4u · 2024-10-14

**Recommendation:** Accept with minor revision

**Comment:**

The conclusion of the reviewers of paper 2153 was the paper proposes an algorithm that is an almost straightforward but potentially useful combination of existing methods. However, they insisted that in order to prove this usefulness, a better discussion of the experimental results was necessary.

Reviewer K8JC of paper 2153 was able to check the new version, under the new pseudonym NqKg. However, Reviewers A9N5 and WWin of paper 2153 were unfortunately not able to review this new version. This is not ideal, but this gave me no choice but to invite two new reviewers, F54M and iDhV. Unfortunately, iDhV him/herself let us down. I thus had to find an emergency reviewer, 9rKf, in the middle of the summer, which led to a very long delay for this paper. I apologize to the authors for this delay, and take the opportunity to thank reviewer 9rKf.

The comments of NqKg are particularly important as this reviewer requested changes on paper 2153. He/she found this new version much better, but still believes that some of the flaws from the previous version remain: some unsupported claims, and lack of clarity in the exposition of the method.

Reviewer F54M was more positive on the readability and praised the authors' effors to include many figure to explain their method. However, he./she also raised point quite close to the ones that were raised for paper 2153. Some of them (lack of novelty, comination of existing ideas) are not part of the decision criteria, but some are quite important, for example, the lack of adequate baselines for comparison in the experiments.

Finally, while his/her opinion seems overall positive, reviewer 9rKf shares the opinion of NqKg (and of the reviewers of paper 2153) about the lack of clarity and the unsupported claims: many reasons are invoked for the use of Bayesian methods in deep learning, but the paper does not provide enough evidence that these promises are fullfilled.

The authors took some time to reply to these comments in depth, and submitted a revised version.

Reviewer 9rKf believes that his/her comments were taken into account seriously: "I appreciate the authors' revisions and response", " The authors' revision now contains a helpful list of the paper's contributions". This leads this reviewer to support publication: "this contributions are fairly subtle, but I now appreciate that the paper's main thrust is in adapting normalizing flow-based inference to LBBNNs." Reviewer NqKg also supports publication, after a minor revision: "I think the presentation (with the exception of two revisions I suggest below) is now adequate." Reviewer F54M, on the other hand, acknowledges that the authors response on the lack of baseline makes sense, but still believes that this makes the results of the paper slightly below the bar for publication. On the other hand, he/she does not oppose publication if it us supported by the other reviewers.

I must say that I share the impression of 9rKf and NqKg: the first version implemented some of the changes requested on paper 2153, but not completely, however the version submitted after the discussion with the authors is now satisfying. I will therefore recommend to accept the paper under minor revision, so that the authors can fix the figures as requested by NqKg.

**Audience:**

Potentially large audience (all the Bayesian ML community, beyond Bayesian deep learning).

**Claims And Evidence:**

This is the resubmission of paper 2153, where the authors proposed a new approach to sparsify Bayesian neural networks learnt. This is typically done with spike-and-slab priors (and posteriors when one uses variational approximations). The authors included latent variables that will enfore some weights to be in the same "state" (spike, or slab), and thus to be simultateously set to 0 or not. This latent structure fits nicely with the normalizing flow used in the algorithm. The algorithm was tested on the MNIST dataset and logistic regression with synthetic data. They claimed that the results match the state of the art in terms of accuracy, with the benefit of uncertainty quantification due to the Bayesian approach. In this new version, the authors improved the discussion of the algorithm and of the experiments as requested by the reviewers.